# 🦕CARINOX: Inference-time Scaling with Category-Aware Reward-based Initial Noise Optimization and Exploration

**Seyed Amir Kasaei**                                    *a.kasaei@me.com*
*Department of Computer Engineering*
*Sharif University of Technology*

**Ali Aghayari**                                    *ali.aghayari@hotmail.com*
*Department of Computer Engineering*
*Sharif University of Technology*

**Arash Marioriyad**                                    *arashmarioriyad@gmail.com*
*Department of Computer Engineering*
*Sharif University of Technology*

**Niki Sepasian**                                    *sepasian.niki@gmail.com*
*Department of Computer Engineering*
*Sharif University of Technology*

**Shayan Baghayi Nejad**                                    *shayanbagha82@gmail.com*
*Department of Computer Engineering*
*Sharif University of Technology*

**MohammadAmin Fazli**                                    *fazli@sharif.edu*
*Department of Computer Engineering*
*Sharif University of Technology*

**Mahdieh Soleymani Baghshah**                                    *soleymani@sharif.edu*
*Department of Computer Engineering*
*Sharif University of Technology*

**Mohammad Hossein Rohban**                                    *rohban@sharif.edu*
*Department of Computer Engineering*
*Sharif University of Technology*

**Reviewed on OpenReview:** *https://openreview.net/forum?id=XB1cwXHVOc*

## Abstract

Text-to-image diffusion models, such as Stable Diffusion, can produce high-quality and diverse images but often fail to achieve *compositional alignment*, particularly when prompts describe complex object relationships, attributes, or spatial arrangements. Recent inference-time approaches address this by optimizing or exploring the *initial noise* under the guidance of reward functions that score text–image alignment—without requiring model fine-tuning. While promising, each strategy has intrinsic limitations when used alone: optimization can stall due to poor initialization or unfavorable search trajectories, whereas exploration may require a prohibitively large number of samples to locate a satisfactory output. Our analysis further shows that neither single reward metrics nor ad-hoc combinations reliably capture all aspects of compositionality, leading to weak or inconsistent guidance. To overcome these challenges, we present **C**ategory-**A**ware **R**eward-based **I**nitial **N**oise **O**ptimization and E**X**ploration (**CARINOX**), a unified framework that combines noise optimization and exploration with a principled reward selection procedure grounded in correlation with

human judgments. Evaluations on two complementary benchmarks—covering diverse compositional challenges—show that **CARINOX** raises average alignment scores by +16% on T2I-CompBench++ and +11% on the HRS benchmark, consistently outperforming state-of-the-art optimization and exploration-based methods across all major categories, while preserving image quality and diversity. The project page is available at this URL.

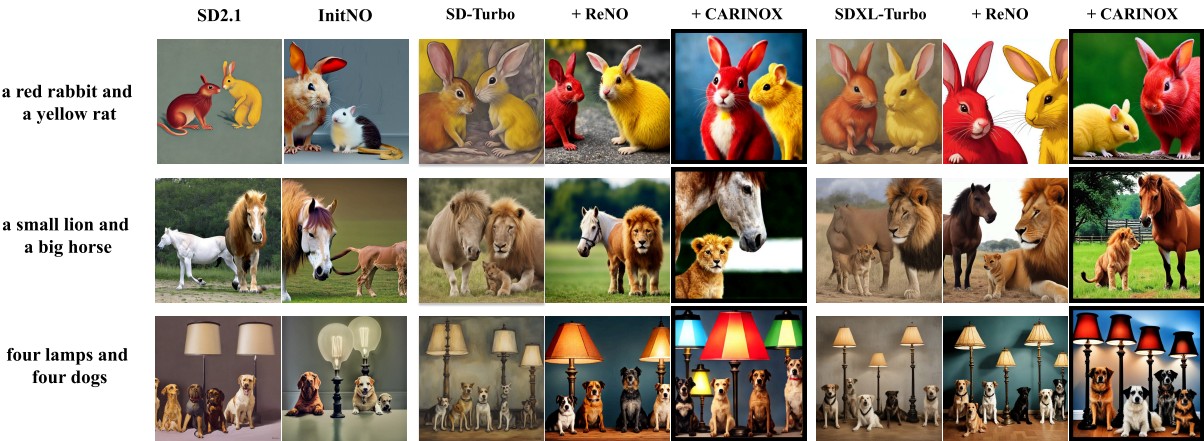

Figure 1: Qualitative results on *T2I-CompBench++*, showing that **CARINOX** faithfully captures compositional details such as counts, spatial arrangements, and attribute bindings.

# 1 Introduction

Text-to-image (T2I) diffusion models, such as Stable Diffusion (SD) (Rombach et al., 2022; Podell et al., 2023) and DALL-E (Ramesh et al., 2022), have garnered substantial attention for their ability to synthesize high-quality images from natural language prompts through iterative denoising and cross-modal attention mechanisms. These models have been adopted in a wide range of applications, including image editing (Huang et al., 2024b; Kawar et al., 2023; Liu et al., 2024a; Mou et al., 2024), data augmentation (Li et al., 2024c; Xiao et al., 2023; Feng et al., 2023a), medical imaging (Huang et al., 2024a; Li et al., 2024b; Khader et al., 2023; Lin et al., 2024a), and marketing (Shilova et al., 2023; Yang et al., 2024). Despite their versatility and impressive generation capabilities, T2I diffusion models often exhibit notable failures in compositional alignment (Huang et al., 2025; Bakr et al., 2023; Ghosh et al., 2024). These failures manifest in various forms, including *entity omission* (Chefer et al., 2023; Sueyoshi & Matsubara, 2024; Zhang et al., 2024; Liu et al., 2022; Kim et al., 2023), *incorrect attribute binding* (Feng et al., 2023b; Singh & Zheng, 2023; Rassin et al., 2024; Wang et al., 2024), *misrepresentation of spatial relationships* (Zhang et al., 2024; Gokhale et al., 2022; Chen et al., 2024), and *numeracy errors* (Binyamin et al., 2024; Zafar et al., 2024; Kang et al., 2023b).

To address compositional generation failures, several studies have explored fine-tuning-based approaches. While effective, such methods are often computationally expensive and time-consuming. In response, a range of inference-time techniques has emerged, aiming to improve generation quality without modifying the underlying model. A similar trend has been observed in large language models (LLMs), where recent work enhances reasoning capabilities by employing verifiers—such as reward functions—during inference rather than through fine-tuning. Within the T2I domain, a subset of inference-time methods focuses on leveraging the initial noise to improve alignment. These approaches fall into two main categories: optimization-based methods, such as ReNO (Eyring et al., 2024) and InitNo (Guo et al., 2024b), which iteratively refine the initial noise to maximize alignment based on a reward signal; and exploration-based methods, including ImageSelect (Karthik et al., 2023), SeedSelect (Samuel et al., 2024b), SemI (Mao et al., 2024), ParticleFiltering (Liu et al., 2024b), and ReliableRandomSeeds (Li et al., 2024a), which evaluate multiple noise samples and select the one yielding the best result. In both settings, reward functions guide the process by scoring how well each candidate image matches the input prompt.

Despite recent progress, existing approaches face two critical challenges that we address in this work. First, both continuous noise optimization and discrete noise selection strategies suffer from inherent limitations when used in isolation. Optimization methods are sensitive to the choice of initial noise and may fail to align the generated image with the prompt due to poor initialization or unfavorable optimization trajectories—even when the starting image appears qualitatively plausible (see Figure 2a). In contrast, exploration-based methods are limited by the nature of their search process: they typically sample from a fixed set of candidates and evaluate each independently, often requiring many trials to find a well-aligned output, particularly in the high-dimensional latent space of diffusion models (see Figure 2b). These limitations are analyzed in more detail in Section 3. Second, the choice of reward function is crucial for guiding generation, yet remains underexplored. Many existing works adopt commonly used metrics without accounting for the specific challenges of compositionality, such as spatial reasoning, entity binding, or numeracy. As a result, the reward signal may be weak or misaligned, reducing the effectiveness of both optimization and exploration.

To overcome these limitations, we propose CARINOX, a novel framework that integrates both noise optimization and exploration strategies with a carefully selected reward function to improve compositional alignment in T2I generation. CARINOX addresses the shortcomings of existing methods by combining continuous optimization of initial noise with a targeted discrete exploration strategy, effectively reducing the risk of poor optimization paths and the inefficiency of blind sampling. To support this process, we systematically derive a robust combination of reward metrics through an empirical correlation study against human judgments, ensuring that the guidance used during generation is aligned with compositional quality. Through this design, CARINOX unifies the strengths of both optimization and exploration while grounding the reward function in a principled, data-driven selection process tailored to compositional challenges.

We evaluate **CARINOX** on two widely used benchmarks—T2I-CompBench++ Huang et al. (2025) and HRS (Bakr et al., 2023)—covering a broad spectrum of compositional challenges. Across both datasets, **CARINOX** consistently improves over the underlying backbones. On *T2I-CompBench++*, it raises the average performance of SD-Turbo from 0.39 to 0.57, SDXL-Turbo from 0.41 to 0.57, and PixArt-$\alpha$ from 0.35 to 0.58, with the strongest gains in texture, numeracy, and spatial reasoning. On the *HRS benchmark*, it further enhances all three backbones, delivering mean improvements of +0.18 on SD-Turbo, +0.16 on SDXL-Turbo, and +0.23 on PixArt-$\alpha$, and setting new highs in creativity, style, and visual writing. Notably, these gains are achieved while preserving image quality and diversity, showing that **CARINOX** strengthens compositional alignment without compromising realism.

## 2 Related Works

Research on compositional generation in T2I diffusion models can be grouped into two families: *fine-tuning methods*, which update model parameters, and *inference-time methods*, which enhance alignment without additional training. Fine-tuning either modifies the denoiser or the text encoder. Denoiser-level updates adapt the UNet or add auxiliary modules for spatial control and attribute binding Sun et al. (2023); Jiang et al. (2024); Guo et al. (2024a); Zhang et al. (2023); Mou et al. (2023), but demand extra compute and risk overfitting. Encoder-level fine-tuning instead adjusts the conditioning space with lightweight projections or causal refinements over frozen CLIP embeddings Zarei et al. (2025), offering better generalization but weaker handling of spatial errors. Inference-time methods operate at different stages of generation. **Prompt-level** rewriting with lexical search or LLM feedback improves attributes and personalization Yu et al. (2024); He et al. (2025), though often costly and verbose. **Embedding-level** adjustments refine frozen encoders to control object counts, attributes, or relations Zafar et al. (2024); Deckers et al. (2024), but are sensitive to hyperparameters. Finally, **noise- and latent-level methods** exploit the strong influence of initialization, forming the basis for optimization and exploration strategies detailed in the next subsections.

### 2.1 Inference-Time Latent-Space Search and Optimization

**Discrete Noise Exploration.** From this category, ImageSelect Karthik et al. (2023) and SeedSelect Samuel et al. (2024b) search over candidate seeds, choosing the one best matching a scoring heuristic (e.g., CLIP similarity). SemI Mao et al. (2024) biases selection toward noise vectors empirically linked to stronger object binding, exploiting "lucky" seeds as reproducible advantages. ParticleFiltering Liu et al. (2024b)

instead performs sequential resampling during reverse diffusion, discarding low-scoring partial generations and retaining promising ones. Relatedly, recent inference-time scaling methods formulate reward alignment as *Sequential Monte Carlo (SMC)* sampling with a population of particles and sequential resampling: DAS Kim et al. (2025b) performs test-time alignment of diffusion models via SMC-based sampling to improve reward while mitigating over-optimization, Kim et al. (2025a) extend particle-based scaling to flow models by injecting stochasticity and reallocating compute via rollover budget forcing, and $\Psi$-Sampler Yoon et al. (2025) improves SMC-based alignment in score models by initializing particles from a reward-aware posterior rather than a Gaussian prior. These methods are fully training-free and turn stochastic seed choice into systematic search, but incur high computational cost, depend on potentially noisy scoring signals, and remain insufficient when base models exhibit severe binding failures.

**Continuous Initial Noise Optimization.** These methods mainly instead refine the initial noise iteratively using reward signals from the final generated image, directly enforcing compositional constraints at test time. InitNO Guo et al. (2024b) optimizes noise with an attention-aware objective that penalizes missing objects and concept mixing, steering sampling away from neglect-inducing regions. ReNO Eyring et al. (2024) extends this to multi-reward optimization, ascending gradients of preference models (e.g., text alignment or detection scores) with respect to noise, improving counting, co-occurrence, and attribute binding. Beyond these, DOODL Wallace et al. (2023) performs end-to-end optimization of the initial diffusion latent by backpropagating losses defined on the final generated image through the denoising process. Related source-latent optimization has also been explored for flow-based generators: D-Flow Ben-Hamu et al. (2024) differentiates through flow-based generation to optimize the source noise under arbitrary costs, and ORIGEN Min et al. (2025) applies reward-guided updates of the initial latent in a one-step flow model to enforce 3D orientation grounding. These methods provide strong, training-free gains and flexibly integrate new constraints, but add inference overhead from iterative scoring/backpropagation, remain sensitive to reward design (risk of reward hacking), and rely on external scorer quality.

**Continuous Latent Optimization.** The category of methods refine noisy latent codes using loss functions on intermediate cross-attention maps, encouraging concept preservation and disentanglement during denoising. Attend-and-Excite Chefer et al. (2023) mitigates neglect by amplifying subject-token activations, while Divide&Bind Li et al. (2023c) adds attendance and binding losses for multi-entity prompts and attribute–object pairing. Predicated Diffusion Sueyoshi & Matsubara (2024) encodes prompt semantics as predicate-logic propositions and treats attention maps as fuzzy predicates, enabling differentiable objectives for complex relations. Attention Regulation Zhang et al. (2024) formulates cross-attention control as constrained optimization that suppresses dominant tokens and boosts under-attended ones, and A-STAR Agarwal et al. (2023) combines attention *segregation* (reducing token overlap) with *retention* (preserving salience across timesteps). Collectively, these training-free methods improve semantic fidelity, recall, and binding by targeting the attention interface, but add inference overhead and are sensitive to hyperparameters balancing faithfulness, diversity, and runtime.

## 2.2 Reward Models for Text-Image Alignment

Compositional alignment rewards assess how well a generated image matches a text prompt in terms of objects, attributes, and spatial relations. Embedding-based methods are widely used, with *CLIPScore* computing similarity between CLIP embeddings of text and image Hessel et al. (2021). Extensions include *HPS*, which fine-tunes CLIP on preference data to better match human judgments Wu et al. (2023), and *PickScore*, which adapts CLIP-H with preference supervision for closer correlation with human rankings Kirstain et al. (2023). Moreover, *ImageReward* trains a standalone reward model on human evaluations to capture prompt relevance and perceptual quality Xu et al. (2024). Complementary to these, VQA-based methods assess alignment by testing whether the image supports answers to prompt-derived questions: *TIFA* generates structured QA pairs and checks them with a pretrained VQA model Hu et al. (2023), while *VQAScore* applies a similar principle and achieves higher correlation with human judgments Lin et al. (2024b). Other approaches in this family, such as DA, DSG, and B-VQA Singh & Zheng (2023); Cho et al. (2023); Huang et al. (2023), also rely on question–answer correctness to provide compositional faithfulness scores.

# 3 Analysis of Optimization and Exploration Limitations

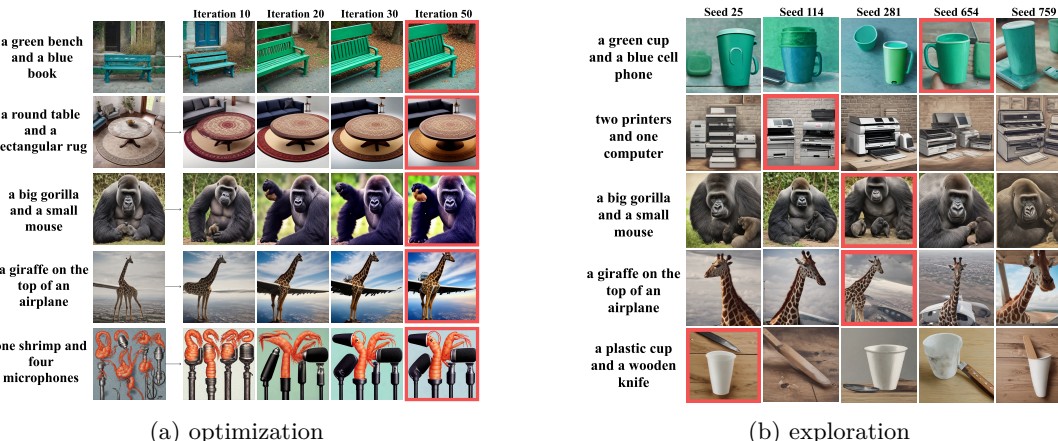

(a) optimization          (b) exploration

Figure 2: Limitations of optimization (a) and exploration (b) when applied in isolation. Optimization often fails to capture attributes or relations despite refinement, while exploration struggles to reliably recover all prompt elements even with multiple seeds.

To analyze the limitations of optimization and exploration in isolation, we conducted controlled experiments with ReNO (Eyring et al., 2024) (optimization) and ImageSelect (Karthik et al., 2023) (exploration), both applied to Stable Diffusion Turbo and guided by a single reward function (ImageReward). A subset of T2I-CompBench++ prompts was used to cover diverse compositional challenges.

**Optimization-based Methods.** Optimization refines a single noise vector but often fails to reach correct compositions when initialization is poor. As shown in Figure 2a, objects may be missing (rows 1 and 3, the blue book and the mouse are absent), attributes may be incorrect (row 2, the rug is generated with a rounded rather than rectangular shape), spatial relations may be wrong (row 4, the giraffe appears in front of rather than on top of the airplane), and counts may be violated (row 5, the correct number of shrimp and microphones is not produced). These cases illustrate how optimization alone can stagnate or diverge, leaving key compositional constraints unsatisfied.

**Exploration-based Methods.** Exploration samples multiple seeds and selects the highest-rewarded output, but due to the sparsity of well-aligned solutions in the noise space, many candidates remain incorrect. As illustrated in Figure 2b, typical failures include missing objects (row 1, no blue cellphone; row 3, missing mouse), incorrect counts (row 2, two printers are generated but the required computer is missing), violated spatial relations (row 4, giraffe misplaced in front of rather than on top of the airplane), and incorrect attributes (row 5, knife texture not faithfully captured). While exploration improves diversity compared to optimization alone, random sampling without refinement rarely guarantees accurate compositional alignment.

Overall, these findings highlight that optimization and exploration in isolation are insufficient for robust compositional alignment, motivating their integration in our unified framework.

# 4 CARINOX: Reward-Guided Initial Noise Optimization and Exploration

We introduce **CARINOX**, a framework that enhances compositional alignment in text-to-image diffusion models through inference-time guidance. The approach integrates two key components: (i) a unified strategy that combines noise exploration with gradient-based noise optimization, and (ii) a correlation-driven selection of reward functions. This design enables CARINOX to more effectively navigate and refine the initial noise space, leading to generations that more reliably capture complex compositional specifications.

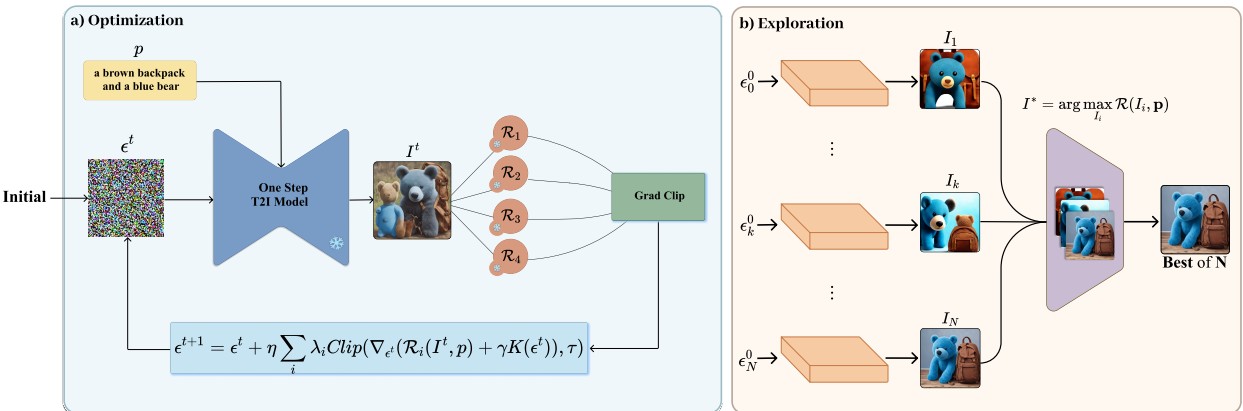

Figure 3: **Overview of the CARINOX framework.** (a) *Optimization*: An initial noise is refined through iterative updates guided by multiple reward functions, with per-reward gradient clipping and latent regularization ensuring stable alignment with the prompt. (b) *Exploration*: Several noise candidates are sampled and independently optimized, and the final image is chosen via best-of-$N$ selection, combining exploration diversity with optimization precision.

## 4.1 Unifying Initial Noise Optimization & Exploration

Improving compositional alignment at inference time can be approached in two ways. *Noise optimization* iteratively refines a single noise vector based on reward signals, while *noise exploration* samples multiple candidates to increase diversity in the search space. Each strategy has clear strengths but also limitations, as discussed in Section 3. In **CARINOX**, we combine these approaches into a *unified* pipeline: exploration broadens the search over initializations, and optimization refines each candidate under a fixed combination of compositional reward functions. The following subsections describe how these components are realized and integrated into our framework.

### 4.1.1 Gradient-Based Initial Noise Optimization

We formulate the optimization of the initial noise vector $\epsilon$ as a *continuous search process* aimed at improving alignment between generated images and textual prompts. Applying gradient-based optimization directly to *multi-step* diffusion models is problematic because the gradient signal must pass through many sequential denoising steps, often leading to vanishing or exploding gradients and significantly increasing computational cost (Eyring et al., 2024). In contrast, *single-step* diffusion models generate the image in one forward pass, allowing gradients from the reward function to propagate cleanly and without degradation. This setup not only eliminates the instability caused by long gradient chains but also makes optimization more efficient, as each iteration requires only one denoising step (see Appendix B for details on single-step models). To further enhance stability and avoid drift into out-of-distribution regions of the latent space, we employ two safeguards: per-reward gradient clipping to prevent any single metric from dominating the update, and latent space regularization to keep $\epsilon$ consistent with the model's prior. These properties make single-step diffusion an ideal choice for our framework, enabling stable and efficient reward-guided refinement of $\epsilon$.

**Noise Refinement via Gradient Ascent**  We aim to refine the initial noise vector $\epsilon$ so that the final generated image aligns more closely with the textual description. The idea is to treat the noise as a set of optimizable parameters, updated iteratively in the direction that increases a reward function measuring text–image alignment.

Formally, given a prompt $\mathbf{p}$ and an initial noise sample $\epsilon$, the generative diffusion model $G_\theta$ maps them to an output image $I$ in a single forward denoising step as $I = G_\theta(\epsilon, \mathbf{p})$.

The quality of this image is evaluated using a composite reward function $\mathcal{R}(I, \mathbf{p})$, which aggregates several pre-selected reward metrics $\mathcal{R}_i$. Each metric measures a different aspect of alignment, such as object cor-

rectness, attribute binding, or spatial relationships. We sum these metric scores to form the optimization objective:

$$\boldsymbol{\epsilon}^* = \arg\max_{\boldsymbol{\epsilon}} \mathcal{R}(I, \mathbf{p}), \tag{1}$$

$$\mathcal{R}(I, \mathbf{p}) = \sum_i \lambda_i \mathcal{R}_i(I, \mathbf{p}). \tag{2}$$

where $\lambda_i$ are the fixed weights assigned to each reward function, set to 1 for all rewards in our implementation.

To adjust $\boldsymbol{\epsilon}$, we compute the gradient of the reward function with respect to the noise vector. This is achieved via the chain rule, first differentiating the reward with respect to the generated image and then propagating this signal backward through the generative model to the noise space:

$$\nabla_{\boldsymbol{\epsilon}} \mathcal{R} = \frac{\partial \mathcal{R}(I, \mathbf{p})}{\partial I} \cdot \frac{\partial I}{\partial \boldsymbol{\epsilon}}. \tag{3}$$

Finally, we update the noise vector using gradient ascent:

$$\boldsymbol{\epsilon}^{(t+1)} = \boldsymbol{\epsilon}^{(t)} + \eta \nabla_{\boldsymbol{\epsilon}} \mathcal{R}, \tag{4}$$

where $\eta$ is the learning rate controlling the step size of the update. This iterative process moves the noise toward configurations that, when decoded by the model, yield images that more closely match the intended compositional structure described in the prompt.

**Gradient Clipping with Multi-Backward Optimization.** Different reward components can produce gradients with vastly different magnitudes, which can destabilize the optimization if one metric dominates the update direction. To address this, we adopt a *multi-backward optimization* strategy, in which the gradient of each reward component is computed separately and clipped before aggregation. This ensures that all metrics contribute in a balanced way, regardless of their natural scale.

Formally, for each reward $\mathcal{R}_i$, the gradient with respect to the noise vector $\boldsymbol{\epsilon}$ is computed as:

$$\nabla_{\boldsymbol{\epsilon}} \mathcal{R}_i = \frac{\partial \mathcal{R}_i(I, \mathbf{p})}{\partial \boldsymbol{\epsilon}}. \tag{5}$$

We then apply $\ell_2$-norm gradient clipping with a maximum norm $\tau = 0.01$. If the gradient's $\ell_2$-norm exceeds $\tau$, it is rescaled proportionally so that:

$$\|\nabla_{\boldsymbol{\epsilon}} \mathcal{R}_i\|_2 \leq \tau. \tag{6}$$

This prevents excessively large updates from any single reward component while preserving their relative direction. After clipping, the gradients from all rewards are aggregated into a single update direction:

$$\nabla_{\boldsymbol{\epsilon}} \mathcal{R} = \sum_i \lambda_i \nabla_{\boldsymbol{\epsilon}} \mathcal{R}_i, \tag{7}$$

where $\lambda_i$ are the fixed weights assigned to each reward function, set to 1 for all rewards in our implementation. We additionally report an ablation on adaptive reward weighting in Appendix F. This procedure ensures that no single reward term overwhelms the optimization, allowing for stable and balanced gradient-based updates (see Appendix E for experimental analysis).

**Regularization for Latent Space Consistency.** During optimization, the noise vector $\boldsymbol{\epsilon}$ can drift far from the distribution it was originally sampled from, namely the standard normal prior $\mathcal{N}(0, I)$. If this happens, the denoiser may receive out-of-distribution inputs, which can degrade image quality and reduce alignment. To prevent such drift, we add a regularization term that encourages $\boldsymbol{\epsilon}$ to remain statistically consistent with the prior distribution.

Following the approach of NAO (Samuel et al., 2024a) and ReNO (Eyring et al., 2024), we do not simply enforce a fixed norm constraint. Instead, we maximize the log-likelihood of the noise vector's norm under

the assumption that it follows a $\chi^d$ distribution (the distribution of the norm of a $d$-dimensional Gaussian vector). This leads to the regularization function:

$$K(\boldsymbol{\epsilon}) = (d - 1)\log(\|\boldsymbol{\epsilon}\|) - \frac{\|\boldsymbol{\epsilon}\|^2}{2}. \tag{8}$$

The final optimization objective combines the main reward function with this regularization term:

$$\mathcal{C} = \mathcal{R}(I, \mathbf{p}) + \gamma K(\boldsymbol{\epsilon}), \tag{9}$$

where $\gamma$ controls the trade-off between maximizing reward and preserving distributional consistency. This regularization constrains the search to noise vectors that are statistically consistent with the model's training distribution, avoiding drift into regions where the denoiser produces unreliable outputs.

### 4.1.2 Noise Exploration for Robust Initialization

Gradient-based optimization is inherently sensitive to the quality of its starting point. If the initial noise vector lies in a region of the latent space that is poorly aligned with the prompt, the optimization process may converge to a suboptimal solution or fail to capture the intended composition entirely. This sensitivity is especially problematic in reward landscapes that are highly non-convex, where poor initialization can trap the optimization in local optima.

We add a *noise exploration* stage that increases the diversity of starting points by drawing $N$ candidates $\{\boldsymbol{\epsilon}_1, \ldots, \boldsymbol{\epsilon}_N\} \sim \mathcal{N}(0, I)$. Each candidate is then refined independently with gradient-based optimization, producing optimized vectors $\{\boldsymbol{\epsilon}_1^*, \ldots, \boldsymbol{\epsilon}_N^*\}$ from which the final output is selected.

**Best-of-N Selection.** After refinement, each optimized noise vector $\boldsymbol{\epsilon}_i^*$ is decoded by the generative model to produce images $\{I_1, \ldots, I_N\}$, where $I_i = G_\theta(\boldsymbol{\epsilon}_i^*, \mathbf{p})$. The final output is selected as the image with the highest composite reward:

$$I^* = \arg\max_{I_i} \mathcal{R}(I_i, \mathbf{p}). \tag{10}$$

In practice, we set $N = 5$ as a balance between efficiency and performance, with ablation results reported in Section 6.1.

This selection strategy offers two key benefits. First, it introduces *diversity through exploration*, ensuring that even if some seeds start far from promising regions, others may lead to stronger alignments. Second, it complements this exploration with *precision through optimization*, as each seed undergoes gradient-based refinement before evaluation. Together, these aspects reduce sensitivity to suboptimal noise initializations and consistently yield high-quality, prompt-consistent results.

### 4.2 Correlation-Guided Reward Combination Selection

Reward functions serve as evaluators in both optimization and exploration, determining whether noise adjustments lead to genuine improvements in text–image alignment. By capturing aspects such as semantic accuracy, attribute binding, and spatial relations, they ensure that optimization emphasizes perceptually meaningful changes. Given their *central role*, reward models must be chosen carefully rather than by ad hoc or popular defaults.

To guide this choice, we conducted a systematic correlation study on the T2I-CompBench++ dataset (Huang et al., 2023), which provides curated prompts, generated images, and human evaluation scores. We tested five embedding-based metrics (PickScore (Kirstain et al., 2023), CLIPScore (Hessel et al., 2021), HPS (Wu et al., 2023), ImageReward (Xu et al., 2024), BLIP-2 (Li et al., 2023a)), five VQA-based metrics (B-VQA (Huang et al., 2023), DA Score (Singh & Zheng, 2023), TIFA (Hu et al., 2023), DSG (Cho et al.,

2023), VQA Score (Lin et al., 2024b)), and two image-only metrics (CLIP-IQA (Wang et al., 2023), Aesthetic Score (Schuhmann et al., 2022)). Spearman rank correlation was used to assess alignment with human preference annotations across compositional categories (Table 5).

The study yielded several insights Kasaei et al. (2025). *No single metric was consistently optimal*: performance varied across attributes, spatial relations, and numeracy. *CLIPScore, despite widespread use, never ranked among the top metrics*, underscoring its weakness as a standalone reward. *VQA-based metrics showed strong compositional reasoning* but were not uniformly superior across categories. Embedding-based metrics such as HPS and ImageReward frequently appeared among the top performers, showing correlations comparable to VQA-based scores. As expected, *image-only metrics correlated poorly* with human judgments.

To construct a reliable reward set, we applied a *top-3 frequency analysis*, counting how often each metric ranked among the three most human-aligned in each category (Table 7). This identified HPS, ImageReward, DA Score, and VQA Score as the most consistently effective. We therefore fix this combination as the unified reward set for CARINOX, ensuring balanced coverage of both *global semantic alignment* and *fine-grained compositional accuracy*. Complete correlation tables and per-category breakdowns are provided in Appendix C.

| Model | Color ↑ | Shape ↑ | Texture ↑ | 2D Spatial ↑ | 3D Spatial ↑ | Numeracy ↑ | Non-Spatial ↑ | Complex ↑ | Mean ↑ |
|---|---|---|---|---|---|---|---|---|---|
| (1) SD-Turbo | 0.47 | 0.37 | 0.45 | 0.48 | 0.39 | 0.45 | 0.59 | 0.46 | 0.46 |
| (1) + ReNO | 0.63 | 0.61 | 0.63 | 0.66 | 0.59 | 0.61 | 0.72 | 0.48 | 0.62 |
| (1) + CARINO | 0.69 | 0.66 | 0.78 | 0.71 | 0.61 | 0.63 | 0.76 | 0.51 | 0.67 |
| (1) + **CARINOX** | 0.78 | 0.74 | **0.89** | **0.80** | 0.65 | **0.71** | **0.80** | 0.62 | 0.75 |
| (2) SDXL-Turbo | 0.61 | 0.61 | 0.75 | 0.69 | 0.69 | 0.44 | 0.71 | 0.47 | 0.62 |
| (2) + ReNO | 0.76 | 0.67 | 0.79 | 0.73 | 0.70 | 0.53 | 0.76 | 0.55 | 0.69 |
| (2) + CARINO | 0.79 | 0.77 | 0.82 | 0.75 | 0.76 | 0.64 | 0.78 | 0.61 | 0.74 |
| (2) + **CARINOX** | **0.86** | **0.85** | 0.87 | 0.78 | **0.78** | 0.66 | 0.79 | **0.71** | **0.79** |

Table 1: Human evaluation on *T2I-CompBench++*, showing that **CARINOX** achieves the highest alignment across categories and backbones. Best values are in bold, second-best are underlined.

# 5 Experiments & Results

## 5.1 Experimental Setup

We evaluate *CARINOX* through a series of experiments designed to assess both compositional alignment and broader generation quality. Human evaluation (Section 5.2) provides direct judgments of alignment quality across multiple backbones. Automated evaluation on *T2I-CompBench++* (Section 5.3) measures performance across diverse compositional categories, while the *HRS benchmark* (Section 5.4) extends this analysis to higher-level aspects such as creativity, style, and visual writing.

For clarity, we compare three variants of our approach: *CARINX*, which applies our fixed reward combination for best-of-$N$ exploration; *CARINO*, which performs initial noise optimization with our reward-guided pipeline; and *CARINOX*, the full method that integrates both exploration and optimization. Together, these benchmarks and variants provide a comprehensive view of how our framework compares to baselines and state-of-the-art (SOTA) methods.

## 5.2 Human-Centered Assessment of Text–Image Alignment

We conducted a human study to directly assess the effectiveness of our proposed variants in improving text–image alignment. A set of 200 prompts covering all eight compositional categories of T2I-CompBench++ was used with two backbones, SD-Turbo and SDXL-Turbo. For each prompt, human annotators rated the generated images on a 0–3 scale, where scores reflected whether all objects were present, attributes were correctly rendered, and relations such as size, color, numeracy, and spatial layout were faithfully captured. Scores were averaged across raters, normalized to [0, 1], and reported in Table 1. Full details of the protocol are provided in Appendix G.

Table 1 reports the results. On SD-Turbo, the baseline achieves a mean score of 0.46, which rises to 0.62 with ReNO. Our initial noise optimization variant, *CARINO*, further improves alignment to 0.67, while the full method, *CARINOX*, reaches 0.75. The largest margins appear in texture and 2D spatial categories, where CARINOX nearly doubles the baseline. On SDXL-Turbo, the mean improves from 0.62 for the backbone to 0.69 with ReNO, 0.74 with CARINO, and 0.79 with CARINOX. Here, the strongest gains occur in color and shape, with additional improvements in 3D spatial reasoning and complex prompts.

Overall, these results confirm that both of our variants enhance human-perceived compositional alignment, with *CARINOX* consistently delivering the most substantial improvements across backbones.

| Model | Color ↑ | Shape ↑ | Texture ↑ | 2D Spatial ↑ | 3D Spatial ↑ | Numeracy ↑ | Non-Spatial ↑ | Complex ↑ | Mean ↑ |
|---|---|---|---|---|---|---|---|---|---|
| SD v1.4 | 0.3765 | 0.3576 | 0.4156 | 0.1246 | 0.3030 | 0.4461 | 0.3079 | 0.3080 | 0.3299 |
| SD v2.1 | 0.5065 | 0.4221 | 0.4922 | 0.1342 | 0.3230 | 0.4579 | 0.3127 | 0.3386 | 0.3734 |
| SDXL | 0.5879 | 0.4687 | 0.5299 | 0.2133 | 0.3566 | 0.4988 | 0.3119 | 0.3237 | 0.4114 |
| PixArt-$\alpha$-ft | 0.6690 | 0.4927 | 0.6477 | 0.2064 | 0.3901 | 0.5058 | 0.3197 | 0.3433 | 0.4468 |
| DALL-E 3 | 0.7785 | 0.6205 | 0.7036 | 0.2865 | 0.3744 | 0.5880 | 0.3003 | 0.3773 | 0.5036 |
| Structured + SD v2.1 | 0.4990 | 0.4218 | 0.4900 | 0.1386 | 0.3224 | 0.4550 | 0.3111 | 0.3355 | 0.3717 |
| Attn-Exct + SD v2.1 | 0.6400 | 0.4517 | 0.5963 | 0.1455 | 0.3222 | 0.4550 | 0.3111 | 0.3355 | 0.4072 |
| InitNO + SD v2.1 | 0.7038 | 0.4694 | 0.5212 | 0.2027 | 0.3524 | 0.4892 | 0.3105 | 0.3574 | 0.4258 |
| (1) SD-Turbo | 0.5252 | 0.4434 | 0.4888 | 0.1881 | 0.3112 | 0.4914 | 0.3095 | 0.3349 | 0.3866 |
| (1) + Pick A Pic | 0.5871 | 0.4842 | 0.5446 | 0.1504 | 0.3559 | 0.5137 | 0.3123 | 0.3768 | 0.4156 |
| (1) + ImageSelect | 0.6800 | 0.5172 | 0.5775 | 0.2317 | 0.3373 | 0.5027 | 0.3136 | 0.3725 | 0.4416 |
| (1) + ReNO | 0.7800 | 0.6200 | 0.7500 | 0.2200 | 0.3800 | 0.5700 | 0.3200 | 0.4800 | 0.5150 |
| (1) + CARINX | 0.7476 | 0.5661 | 0.6216 | 0.2366 | 0.3421 | 0.5295 | 0.3126 | 0.3841 | 0.4675 |
| (1) + CARINO | 0.8519 | 0.7336 | 0.8043 | 0.2437 | 0.3920 | 0.5903 | 0.3269 | 0.4906 | 0.5542 |
| **(1) + CARINOX** | 0.8633 | 0.7609 | 0.8229 | 0.2588 | 0.4155 | 0.6248 | **0.3372** | **0.5041** | 0.5734 |
| (2) SDXL-Turbo | 0.5959 | 0.4038 | 0.5472 | 0.2303 | 0.3612 | 0.4863 | 0.3114 | 0.3430 | 0.4099 |
| (2) + Pick A Pic | 0.6532 | 0.4803 | 0.6176 | 0.2679 | 0.3959 | 0.5492 | 0.3122 | 0.3741 | 0.4563 |
| (2) + ImageSelect | 0.7369 | 0.5257 | 0.6590 | 0.2426 | 0.3838 | 0.5398 | 0.3115 | 0.3802 | 0.4724 |
| (2) + ReNO | 0.7800 | 0.6000 | 0.7400 | 0.2600 | 0.3900 | 0.5600 | 0.3100 | 0.4700 | 0.5137 |
| (2) + CARINX | 0.7890 | 0.5708 | 0.7068 | 0.2663 | 0.3984 | 0.5423 | 0.3129 | 0.3932 | 0.4975 |
| (2) + CARINO | 0.8492 | 0.7203 | 0.7977 | 0.2858 | 0.4069 | 0.5835 | 0.3141 | 0.4859 | 0.5554 |
| **(2) + CARINOX** | **0.8697** | 0.7482 | **0.8270** | 0.3010 | 0.4117 | 0.5992 | 0.3232 | 0.4922 | 0.5715 |
| (3) PixArt-$\alpha$ DMD | 0.4145 | 0.3487 | 0.3667 | 0.2213 | 0.3441 | 0.4896 | 0.3061 | 0.3466 | 0.3547 |
| (3) + Pick A Pic | 0.4475 | 0.3690 | 0.4658 | 0.1987 | 0.3704 | 0.5393 | 0.3082 | 0.3555 | 0.3818 |
| (3) + ImageSelect | 0.5361 | 0.4406 | 0.5148 | 0.1878 | 0.3747 | 0.5453 | 0.3091 | 0.3634 | 0.4090 |
| (3) + ReNO | 0.6400 | 0.5700 | 0.7200 | 0.2500 | 0.3900 | 0.5600 | 0.3100 | 0.4600 | 0.4875 |
| (3) + CARINX | 0.5966 | 0.4855 | 0.5643 | 0.2348 | 0.3697 | 0.5463 | 0.3100 | 0.3805 | 0.4360 |
| (3) + CARINO | 0.8260 | 0.7528 | 0.7967 | 0.2620 | 0.4031 | 0.6144 | 0.3146 | 0.4782 | 0.5560 |
| **(3) + CARINOX** | 0.8545 | **0.7721** | 0.8076 | **0.3272** | **0.4164** | **0.6295** | 0.3256 | 0.4878 | **0.5776** |

Table 2: Quantitative evaluation on T2I-CompBench++ across eight compositional categories using three different backbones. Results are reported for baseline models, SOTA methods, and our variants. **CARINOX** achieves the strongest overall alignment, consistently surpassing both optimization- and exploration-based baselines. Best values are in bold, and second-best are underlined.

## 5.3 Category-Level Compositional Benchmarking — T2I-CompBench++

We further benchmark our approach against a broad set of SOTA methods on T2I-CompBench++, which evaluates eight compositional categories using specialized evaluators for each dimension. Since different metrics are used across categories (e.g., CLIP for non-spatial relations), the absolute ranges vary, but the comparisons remain consistent across methods. Our baselines cover multiple methodological families: (i) multi-step open-source diffusion backbones (SD v1.4, SD v2.1, SDXL, PixArt-$\alpha$), (ii) *single-step* backbones (SD-Turbo, SDXL-Turbo, PixArt-$\alpha$ DMD), (iii) recent commercial systems such as DALL · E 3, (iv) attention-based inference-time control methods (Structured Diffusion Feng et al. (2023b), Attend-and-Excite Chefer et al. (2023)), (v) gradient-based initial noise optimization approaches (InitNO Guo et al. (2024b), ReNO Eyring et al. (2024)), and (vi) exploration strategies (Pick-a-Pic Kirstain et al. (2023), ImageSelect Karthik et al. (2023)). We report results for three variants of our method: CARINX (exploration only), CARINO (optimization only), and CARINOX (combined).

As summarized in Table 2, CARINOX consistently achieves the highest overall performance across all three backbones. On SD-Turbo, it improves the mean score from 0.39 to 0.57, outperforming ReNO (0.52) and

significantly surpassing exploration-based methods such as Pick-a-Pic (0.42) and ImageSelect (0.44). The gains are most pronounced in texture and numeracy, while stable improvements are also observed in 2D and 3D spatial reasoning. On SDXL-Turbo, CARINOX raises the mean from 0.41 to 0.57, again outperforming all baselines, with particularly strong results in texture and complex categories. Finally, on PixArt-$\alpha$, CARINOX achieves the highest mean score of 0.58, with notable advantages in 2D spatial (0.33 vs. 0.22 for the backbone) and numeracy (0.63 vs. 0.49).

Importantly, CARINO and CARINX also provide consistent improvements when used independently: CARINX surpasses existing exploration methods, while CARINO establishes a new strong baseline for noise optimization. Together, the full CARINOX pipeline further amplifies these gains, outperforming commercial systems such as DALL · E 3 and advancing the SOTA across compositional categories.

| Model | Creativity ↑ | Style ↑ | Size ↑ | Visual Writing ↑ | Mean ↑ |
|---|---|---|---|---|---|
| SD-Turbo | 0.4914 | 0.2370 | 0.2118 | 0.1890 | 0.2823 |
| (1) + Pick A Pic | 0.4950 | 0.2682 | 0.2414 | 0.1974 | 0.3005 |
| (1) + ImageSelect | 0.5319 | 0.2834 | 0.2483 | 0.1872 | 0.3127 |
| (1) + ReNO | 0.5333 | 0.3407 | 0.2614 | 0.2838 | 0.3548 |
| (1) + CARINX | 0.5354 | 0.3033 | 0.2582 | 0.2025 | 0.3249 |
| (1) + CARINO | 0.6105 | 0.4647 | 0.2634 | 0.3739 | 0.4281 |
| **(1) + CARINOX** | **0.6246** | **0.4975** | 0.3006 | **0.4329** | **0.4639** |
| SDXL-Turbo | 0.5093 | 0.2526 | 0.2443 | 0.2569 | 0.3158 |
| (2) + Pick A Pic | 0.5154 | 0.2810 | 0.2620 | 0.2965 | 0.3387 |
| (2) + ImageSelect | 0.5298 | 0.3106 | 0.2696 | 0.3074 | 0.3544 |
| (2) + ReNO | 0.5451 | 0.3691 | 0.2567 | 0.3273 | 0.3746 |
| (2) + CARINX | 0.5326 | 0.3406 | 0.2725 | 0.3215 | 0.3668 |
| (2) + CARINO | 0.5913 | 0.4502 | 0.2704 | 0.3980 | 0.4275 |
| **(2) + CARINOX** | **0.6248** | **0.4907** | **0.3070** | **0.4699** | 0.4731 |
| PixArt-$\alpha$ DMD | 0.4775 | 0.2552 | 0.1677 | 0.1136 | 0.2535 |
| + Pick A Pic | 0.5150 | 0.2749 | 0.1967 | 0.1265 | 0.2783 |
| + ImageSelect | 0.5026 | 0.2832 | 0.1985 | 0.1412 | 0.2814 |
| + ReNO | 0.5445 | 0.3659 | 0.1982 | 0.1975 | 0.3265 |
| + CARINX | 0.5289 | 0.2938 | 0.1993 | 0.1577 | 0.2949 |
| + CARINO | 0.6431 | 0.5076 | 0.2565 | 0.4112 | 0.4546 |
| **+ CARINOX** | **0.6697** | **0.5358** | 0.2849 | 0.4485 | **0.4847** |

Table 3: Evaluation on the *HRS benchmark* across three backbones and variants of our exploration and optimization methods. **CARINOX** achieves the strongest overall performance, with best scores in bold and second-best underlined.

### 5.4 Beyond Compositionality: Expressive Evaluation on HRS

The HRS benchmark extends evaluation beyond strict compositionality to creativity, artistic style, object size, and visual writing. These dimensions test whether a model can balance alignment with expressiveness and stylistic fidelity.

Table 3 shows that **CARINOX** consistently outperforms both the backbones and competing methods. On SD-Turbo, it raises the mean score from 0.28 to 0.46, mainly through large gains in creativity and visual writing, where baseline models are especially weak. On SDXL-Turbo, CARINOX delivers the strongest overall balance, improving all four dimensions simultaneously and setting new best results in size and visual writing. On PixArt-$\alpha$, it again achieves the top mean (0.48), driven by clear advantages in creativity and style while also improving visual writing.

These results demonstrate that CARINOX is not only effective at resolving compositional failures but also enhances higher-level aspects of generation such as artistic quality and written content. Importantly, CARINO and CARINX each provide gains on their own, but their integration in CARINOX consistently produces the most robust improvements.

### 5.5 Qualitative Results

Figures 1 and 4 illustrate that baseline models (SD2.1, InitNO, SD-Turbo, SDXL-Turbo) often miss core compositional requirements, and while ReNO improves alignment, it still produces frequent errors. In contrast, *CARINOX* consistently generates images that better match prompts across diverse settings: on *T2I-CompBench++* it respects relative sizes, attributes, and counts (e.g., "a dog smaller than a chair," "four lamps and four dogs"), while on *HRS* it produces clearer text rendering, coherent styles, and more expressive compositions. These results highlight the robustness of *CARINOX* over both baselines and ReNO.

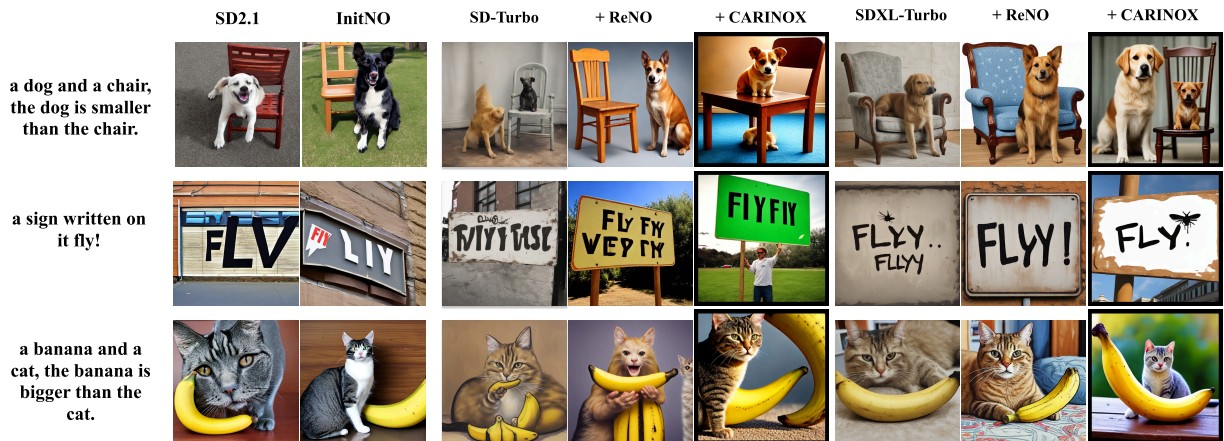

Figure 4: Qualitative results on the *HRS benchmark*, where **CARINOX** produces coherent, visually expressive outputs with accurate style and text rendering.

## 6 Ablation

### 6.1 Effect of Iterations and Seeds

Figure 5 presents the ablation study on the number of optimization iterations and seeds. Increasing the number of iterations improves alignment scores consistently up to about 50, after which the gains plateau and in some categories even decline slightly. Similarly, increasing the number of seeds enhances performance by enlarging the exploration space, but the benefit saturates beyond roughly 5 seeds.

Based on these results, we set the default configuration of CARINOX to *50 optimization iterations* and *5 seeds*. This choice provides an effective balance between computational efficiency and alignment performance, capturing most of the achievable improvement without incurring unnecessary cost.

### 6.2 Evaluation of Individual Reward Functions

To better understand the contribution of each reward function, we analyze the effect of applying them individually within our noise optimization pipeline using SD-Turbo on T2I-CompBench++. We consider four reward models: HPS, ImageReward, DA Score, and VQA Score. Each component is applied separately to guide optimization, allowing us to assess its influence on different compositional categories.

The results in Table 4 indicate that the rewards complement each other rather than excelling universally. DA Score achieves solid improvements in color and shape but is less consistent across other categories. ImageReward provides balanced gains, performing well in texture and spatial reasoning. HPS is particularly effective in numeracy and 2D spatial categories, while VQA Score contributes moderately but lags behind the other metrics in overall mean performance. Crucially, integrating all four into CARINO yields the highest mean score (0.55), a 0.15 improvement over the SD-Turbo baseline and above any single reward.

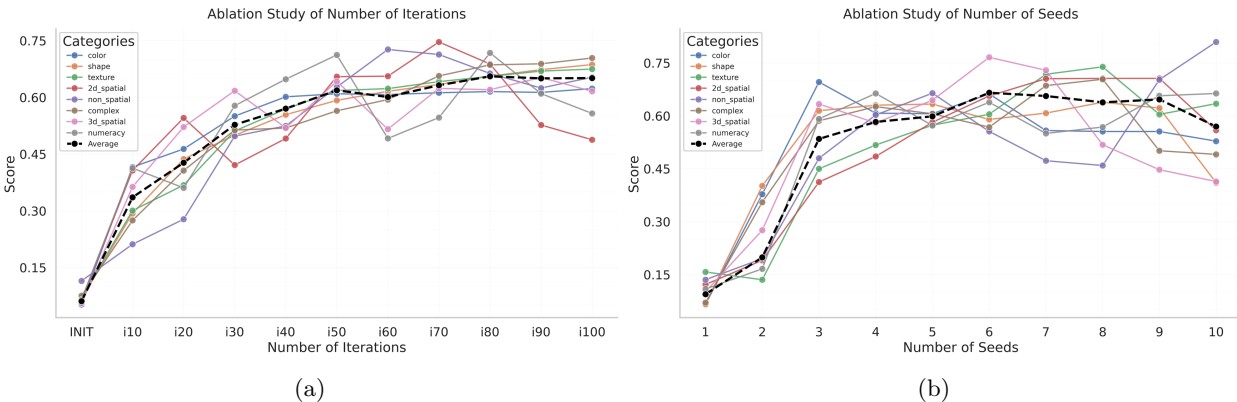

(a)                      (b)

Figure 5: Effect of optimization iterations (a) and exploration seeds (b) on T2I-CompBench++. Performance improves with more iterations and seeds but saturates beyond 50 iterations and 5 seeds, motivating their use as CARINOX defaults for balanced efficiency and alignment.

These results confirm that no individual reward is sufficient on its own, and that strategically combining complementary signals leads to stronger and more reliable alignment improvements across compositional challenges.

| Method | Color ↑ | Shape ↑ | Texture ↑ | 2D Spatial ↑ | 3D Spatial ↑ | Numeracy ↑ | Non-Spatial ↑ | Complex ↑ | Mean ↑ |
|---|---|---|---|---|---|---|---|---|---|
| SD-Turbo | 0.55 | 0.44 | 0.57 | 0.17 | 0.30 | 0.49 | 0.30 | 0.41 | 0.40 |
| + HPS | 0.69 | 0.60 | 0.71 | **0.27** | **0.40** | **0.61** | 0.30 | 0.41 | 0.50 |
| + ImageReward | 0.80 | 0.63 | 0.72 | 0.20 | 0.39 | 0.60 | 0.31 | 0.44 | 0.51 |
| + DA Score | **0.86** | **0.81** | 0.79 | 0.23 | 0.31 | 0.53 | 0.30 | 0.45 | 0.53 |
| + VQA Score | 0.70 | 0.53 | 0.67 | 0.24 | 0.35 | 0.59 | 0.30 | 0.40 | 0.47 |
| + CARINO | 0.85 | 0.73 | **0.80** | 0.24 | 0.39 | 0.59 | **0.33** | **0.49** | **0.55** |

Table 4: Effect of individual reward components on SD-Turbo evaluated on *T2I-CompBench++*. Single rewards might get category-specific gains, while the full CARINO combination achieves the best overall mean and the most balanced improvements across categories.

## 7 Conclusion

We presented **CARINOX**, an inference-time framework that unifies initial noise exploration with gradient-based optimization to improve compositional text-to-image generation. By refining multiple seeds in parallel and selecting the best candidate through a correlation-guided reward combination, CARINOX effectively balances diversity with precision. The framework incorporates gradient clipping to prevent reward dominance and latent regularization to maintain distributional consistency, enabling stable refinement without sacrificing realism. Extensive experiments on **T2I-CompBench++** and **HRS** demonstrate that CARINOX consistently outperforms baselines and prior inference-time approaches, achieving more reliable compositional alignment and higher perceptual quality. These results underscore the potential of optimizing initial noise as a scalable path toward robust inference-time scaling for diffusion models.

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

# Appendix

## A   Future Work

While **CARINOX** demonstrates strong improvements in compositional alignment, several directions remain open for exploration.

First, although we focused on a carefully selected set of reward functions, future work may incorporate *richer or domain-specific reward models*, including those trained on human preference datasets beyond compositional alignment, or multimodal evaluators capable of handling more abstract properties such as style and creativity.

Second, CARINOX currently applies reward feedback within a one-step generative backbone. Extending the framework to *multi-step diffusion models* would allow gradients to propagate across the full denoising trajectory, potentially unlocking finer-grained control over alignment.

Finally, we envision combining advanced reward models with our exploration–optimization pipeline in a more general *reinforcement learning–style framework*, where both reward definitions and update strategies co-evolve to optimize compositional alignment. Together, these directions could make CARINOX not only more robust but also more broadly applicable to future generations of text-to-image systems.

## B   Preliminaries: One-Step Diffusion Models

Diffusion models have become a fundamental approach for text-to-image (T2I) generation, leveraging a stochastic denoising process to progressively refine an initial noise sample into a coherent image (Ho et al., 2020; Rombach et al., 2022). Given a textual prompt $\mathbf{p}$, a diffusion-based generative model $G_\theta$, parameterized by $\theta$, synthesizes an image $\mathbf{x_0}$ by starting from a sampled noise $\mathbf{z}_0 \sim \mathcal{N}(0, I)$ and applying a learned transformation such that:

$$G_\theta(\mathbf{z}_0, \mathbf{p}) = \mathbf{x_0}. \tag{11}$$

The goal of training is to optimize $\theta$ such that the generated image $\mathbf{x_0}$ is semantically aligned with $\mathbf{p}$.

### B.1   From Multi-Step to One-Step Diffusion

Standard diffusion models follow a multi-step denoising process, where an image $\mathbf{x_t}$ at timestep $t$ follows the stochastic transition:

$$\mathbf{x_t} = \alpha_t \mathbf{x_0} + \sigma_t \mathbf{z_0}, \quad t \in [0, T], \tag{12}$$

where $\alpha_t$ and $\sigma_t$ are time-dependent scaling factors such that $\alpha_t$ decreases while $\sigma_t$ increases over time. The reverse process reconstructs $\mathbf{x_0}$ by progressively removing noise through a learned score function. However, this stepwise reconstruction makes inference computationally expensive.

To mitigate this, one-step diffusion models aim to approximate the full denoising trajectory in a single function evaluation by learning a direct mapping from the initial noise to the final image:

$$\mathbf{x_0} = f_\theta(\mathbf{z}_0, \mathbf{p}). \tag{13}$$

This transformation eliminates the need for iterative refinement, significantly reducing inference time while maintaining generative quality.

### B.2   Training and Optimization

One-step diffusion models are typically trained by distilling the multi-step diffusion process into a single-step model. This involves minimizing a reconstruction loss that ensures $f_\theta$ approximates the multi-step generative process:

$$\mathcal{L}(\theta) = \mathbb{E}_{\mathbf{x_0} \sim p(\mathbf{x_0}), \mathbf{z}_0 \sim \mathcal{N}(0, I)} \left[ ||f_\theta(\mathbf{z}_0, \mathbf{p}) - \mathbf{x_0}||^2 \right]. \tag{14}$$

This objective encourages $f_\theta$ to reconstruct high-quality images directly from noise while preserving the semantic content dictated by $\mathbf{p}$.

One-step diffusion models provide an efficient framework for direct noise optimization. By enabling gradient-based refinements of $\mathbf{z}_0$, they serve as the foundation for our proposed reward-driven initial noise optimization framework, which is described in the following section.

## C  Correlation Study of Evaluation Metrics

### C.1  Evaluation Metrics

A range of metrics have been proposed for evaluating text–image alignment, each targeting different aspects of the correspondence. They can be grouped into three categories: (1) *embedding-based*, which rely on representations or preference models; (2) *content-based*, which use structured reasoning to assess compositional properties; and (3) *image-only*, which measure perceptual quality independently of text.

| Metric | Color | Shape | Texture | 2D Spatial | Non-Spatial | Complex | 3D Spatial | Numeracy |
|---|---|---|---|---|---|---|---|---|
| CLIP (Hessel et al., 2021) | 0.282 | 0.291 | 0.535 | 0.369 | 0.439 | 0.276 | 0.315 | 0.223 |
| PickScore (Kirstain et al., 2023) | 0.263 | 0.270 | 0.516 | 0.299 | 0.432 | 0.167 | 0.139 | 0.337 |
| HPS (Wu et al., 2023) | 0.219 | 0.440 | 0.601 | 0.410 | **0.535** | 0.270 | 0.416 | 0.471 |
| ImageReward (Xu et al., 2024) | 0.580 | **0.520** | **0.734** | 0.394 | 0.512 | 0.424 | 0.401 | 0.484 |
| BLIP2 (Li et al., 2023a) | 0.250 | 0.287 | 0.546 | 0.369 | 0.353 | 0.235 | 0.416 | 0.366 |
| Aesthetic (Schuhmann et al., 2022) | 0.056 | 0.195 | 0.078 | 0.136 | 0.061 | 0.051 | 0.123 | 0.036 |
| CLIP-IQA (Wang et al., 2023) | 0.092 | 0.078 | -0.001 | 0.088 | 0.082 | 0.027 | 0.098 | 0.068 |
| B-VQA (Huang et al., 2023) | 0.610 | 0.388 | 0.690 | 0.255 | 0.371 | 0.372 | 0.330 | 0.444 |
| DA Score (Singh & Zheng, 2023) | **0.772** | 0.463 | 0.711 | 0.318 | 0.453 | 0.488 | 0.297 | 0.462 |
| TIFA (Hu et al., 2023) | 0.684 | 0.336 | 0.423 | 0.311 | 0.351 | 0.519 | 0.195 | **0.526** |
| DSG (Cho et al., 2023) | 0.599 | 0.388 | 0.628 | 0.328 | 0.470 | 0.411 | **0.427** | 0.469 |
| VQA Score (Lin et al., 2024b) | 0.678 | 0.405 | 0.701 | **0.533** | 0.495 | **0.638** | 0.339 | 0.473 |

Table 5: Spearman correlation of evaluation metrics with human scores across compositional categories on T2I-CompBench++. The highest value in each category is shown in bold, and the second-highest is underlined.

| Metric | Color | Shape | Texture | 2D Spatial | Non-Spatial | Complex | 3D Spatial | Numeracy |
|---|---|---|---|---|---|---|---|---|
| CLIP (Hessel et al., 2021) | 0.208 | 0.211 | 0.392 | 0.287 | 0.347 | 0.201 | 0.224 | 0.154 |
| PickScore (Kirstain et al., 2023) | 0.193 | 0.192 | 0.373 | 0.229 | 0.341 | 0.122 | 0.100 | 0.241 |
| HPS (Wu et al., 2023) | 0.157 | 0.326 | 0.441 | 0.315 | **0.428** | 0.201 | 0.305 | 0.346 |
| ImageReward (Xu et al., 2024) | 0.434 | **0.388** | **0.549** | 0.310 | 0.408 | 0.313 | 0.294 | 0.349 |
| BLIP2 (Li et al., 2023a) | 0.179 | 0.203 | 0.389 | 0.286 | 0.280 | 0.170 | 0.303 | 0.264 |
| Aesthetic (Schuhmann et al., 2022) | 0.039 | 0.138 | 0.054 | 0.104 | 0.047 | 0.037 | 0.083 | 0.026 |
| CLIP-IQA (Wang et al., 2023) | 0.065 | 0.055 | -0.002 | 0.068 | 0.063 | 0.018 | 0.068 | 0.045 |
| B-VQA (Huang et al., 2023) | 0.456 | 0.279 | 0.512 | 0.195 | 0.293 | 0.267 | 0.231 | 0.322 |
| DA Score (Singh & Zheng, 2023) | **0.603** | 0.337 | 0.534 | 0.247 | 0.357 | 0.364 | 0.206 | 0.347 |
| TIFA (Hu et al., 2023) | 0.559 | 0.246 | 0.329 | 0.266 | 0.292 | 0.405 | 0.155 | **0.400** |
| DSG (Cho et al., 2023) | 0.499 | 0.303 | 0.503 | 0.292 | 0.408 | 0.325 | **0.355** | 0.363 |
| VQA Score (Lin et al., 2024b) | 0.512 | 0.292 | 0.516 | **0.422** | 0.390 | **0.481** | 0.243 | 0.352 |

Table 6: Kendall's $\tau$ correlation of evaluation metrics with human scores across compositional categories on T2I-CompBench++. The highest value in each category is shown in bold, and the second-highest is underlined.

**Embedding-based Metrics**  Embedding-based metrics evaluate alignment by comparing text–image representations in a shared multimodal space or by leveraging models trained on human preferences. A common baseline is *CLIPScore* (Hessel et al., 2021), which measures cosine similarity between CLIP embeddings. Preference-supervised variants include *HPS* (Wu et al., 2023), which fine-tunes CLIP on human comparisons, and *PickScore* (Kirstain et al., 2023), which learns from pairwise preference judgments. *BLIP* (Li et al., 2023a) follows the embedding-similarity approach, comparing captions generated from images with the input text. Extending this idea, *ImageReward* (Xu et al., 2024) adds a reward head trained on ranked human preference data, capturing both textual relevance and perceptual quality.

| Metric | Color | Shape | Texture | 2D Spatial | Non-Spatial | Complex | 3D Spatial | Numeracy | Total |
|---|---|---|---|---|---|---|---|---|---|
| CLIP (Hessel et al., 2021) | | | | | | | | | 0 |
| PickScore (Kirstain et al., 2023) | | | | | | | | | 0 |
| HPS (Wu et al., 2023) | | ✓ | | ✓ | ✓ | | ✓ | | **4** |
| ImageReward (Xu et al., 2024) | | ✓ | ✓ | ✓ | ✓ | | ✓ | ✓ | **6** |
| BLIP2 (Li et al., 2023a) | | | | | | | ✓ | | 1 |
| Aesthetic (Schuhmann et al., 2022) | | | | | | | | | 0 |
| CLIP-IQA (Wang et al., 2023) | | | | | | | | | 0 |
| B-VQA (Huang et al., 2023) | | | | | | | | | 0 |
| DA Score (Singh & Zheng, 2023) | ✓ | ✓ | ✓ | | | ✓ | | | **4** |
| TIFA (Hu et al., 2023) | ✓ | | | | | ✓ | | ✓ | 3 |
| DSG (Cho et al., 2023) | | | | | | | ✓ | | 1 |
| VQA Score (Lin et al., 2024b) | ✓ | | ✓ | ✓ | ✓ | ✓ | | ✓ | **6** |

Table 7: Top-3 presence of each metric across various compositional categories. A ✓ indicates the metric is among the top 3 in that category. The last column shows the total number of categories where the metric appears in the top 3 based on spearman correlation.

**Content-based (VQA-based) Metrics**  VQA-based metrics assess compositional alignment by casting text–image consistency as a question answering task. Questions derived from the prompt are posed to a pretrained VQA model, with scores based on the correctness of its responses. *VQAScore* (Lin et al., 2024b) generates yes/no questions from the text, while *TIFA* (Hu et al., 2023) uses structured templates to cover objects, attributes, and relations. Variants target specific aspects: *DA Score* (Singh & Zheng, 2023) asks entity–attribute questions to test binding, *DSG* (Cho et al., 2023) converts the text into a scene graph to verify entities and relations, and *B-VQA* (Huang et al., 2023) decomposes the text into object–attribute pairs, querying each with BLIP-VQA and combining the probabilities.

**Image-only Metrics**  Image-only metrics assess perceptual quality independently of the prompt, providing complementary signals of realism and aesthetics. *CLIP-IQA* (Wang et al., 2023) predicts image quality by regressing CLIP embeddings against human quality annotations, while the *Aesthetic Score* (Schuhmann et al., 2022) estimates aesthetic value from large-scale human ratings.

## C.2  Experimental Setting

Our analysis is based on T2I-CompBench++ (Huang et al., 2025), which provides curated prompts across attributes (color, shape, texture), spatial relations (2D and 3D), non-spatial relations, complex prompts, and numeracy. Each prompt is paired with images from multiple text-to-image models and annotated with human evaluation scores. All resources (prompts, images, and scores) come from the benchmark; our contribution is to analyze how evaluation metrics align with these annotations using outputs from SD v1.4, SD v2, Structured Diffusion (Feng et al., 2023b), Composable Diffusion (Liu et al., 2022), Attend-and-Excite (Chefer et al., 2023), and GORS (Huang et al., 2023).

We evaluate five embedding-based metrics (PickScore (Kirstain et al., 2023), CLIPScore (Hessel et al., 2021), HPS (Wu et al., 2023), ImageReward (Xu et al., 2024), BLIP-2 (Li et al., 2023a)), two image-only metrics (CLIP-IQA (Wang et al., 2023), Aesthetic Score (Schuhmann et al., 2022)), and five VQA-based metrics (B-VQA (Huang et al., 2023), DA Score (Singh & Zheng, 2023), TIFA (Hu et al., 2023), DSG (Cho et al., 2023), VQA Score (Lin et al., 2024b)), covering embedding similarity, perceptual quality, and VQA-style reasoning.

## C.3  Correlation Analysis of Evaluation Metrics

We assess the reliability of reward models by correlating their scores on T2I-CompBench++ generations with human evaluations (Section C.2). Spearman correlations, reported in Table 5, serve as the main measure, while Kendall's $\tau$ results are provided in Table 6 for completeness.

**Per-Category Breakdown of Correlation Results**   Table 5 highlights that the strongest correlations differ substantially across categories, indicating that *no single metric dominates overall*. In the attribute group, DA Score leads on color (TIFA second), while ImageReward ranks highest on shape and texture (DA Score second). For relational cases, VQA Score performs best on 2D spatial (HPS second), whereas DSG leads in 3D spatial (HPS and BLIP-2 second). Non-spatial relations are best captured by HPS, followed by ImageReward. In complex prompts, VQA Score shows the strongest alignment, with TIFA second, and in numeracy, TIFA ranks first with ImageReward next. *Across all categories, image-only metrics (CLIP-IQA, Aesthetic) remain consistently weak*, underscoring their limited value for compositional alignment.

**Broader Insights on Metric Performance**   Several broader insights emerge from these results. First, *no single metric achieves strong and consistent correlation across all compositional categories*, indicating that reliance on a single signal is insufficient. Second, despite its widespread use (Rombach et al., 2022; Nichol et al., 2021; Ruiz et al., 2023; Brooks et al., 2023; Kumari et al., 2023; Kang et al., 2023a; Chefer et al., 2023; Podell et al., 2023; Chen et al., 2023; Li et al., 2023b; Nguyen & Tran, 2024), *CLIP never ranks among the top metrics*, underscoring its limitations as a standalone measure. Third, embedding-based metrics, particularly ImageReward and HPS, frequently appear among the strongest. Fourth, while VQA-based metrics are competitive, *they are not uniformly superior and are occasionally outperformed by embedding-based approaches*. Finally, image-only metrics such as CLIP-IQA and Aesthetic remain consistently weak, as expected since they do not assess text–image alignment.

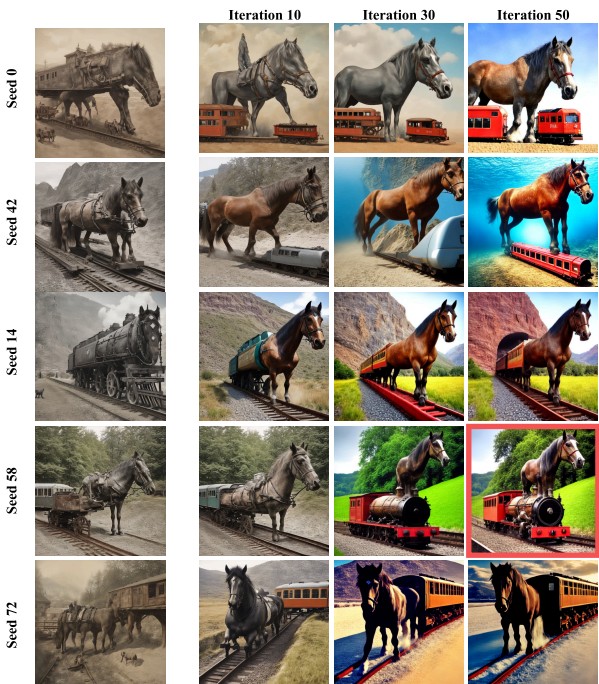

Figure 6: Iterative refinement for the prompt "a train on the bottom of a horse." Five different seeds are optimized in parallel, and by iteration 50, outputs converge toward coherent compositions. The best image is then selected using our reward scores.

# D   Results of Iterative Noise Refinement

Figure 6 illustrates how the proposed noise refinement progressively improves alignment between the generated images and the input prompt. Starting from diverse initial seeds, the early iterations often produce incomplete or ambiguous compositions. As optimization advances, the structure of the scene becomes clearer: the horse and train appear more consistently, their spatial relations stabilize, and extraneous artifacts are

reduced. By iteration 50, the outputs across different seeds converge toward coherent and faithful realizations of the prompt, while still preserving diversity in style and background.

In practice, the framework generates multiple refined candidates in parallel and selects the best image using our reward combination. This best-of-$N$ selection ensures that the final output not only reflects consistent alignment but also represents the strongest candidate among diverse refinements.

# E    Effect of Multi-Clip on Multi-Backward Optimization

In our optimization pipeline, each reward metric contributes gradients that guide noise refinement. However, the magnitudes of these gradients can vary drastically. Without proper regulation, a dominant reward can overpower the others, pushing the optimization toward solutions that satisfy alignment objectives but sacrifice realism. To address this, we apply **Multi-Clip**: a mechanism that clips the gradient of each reward independently before aggregation, ensuring balanced updates across all metrics.

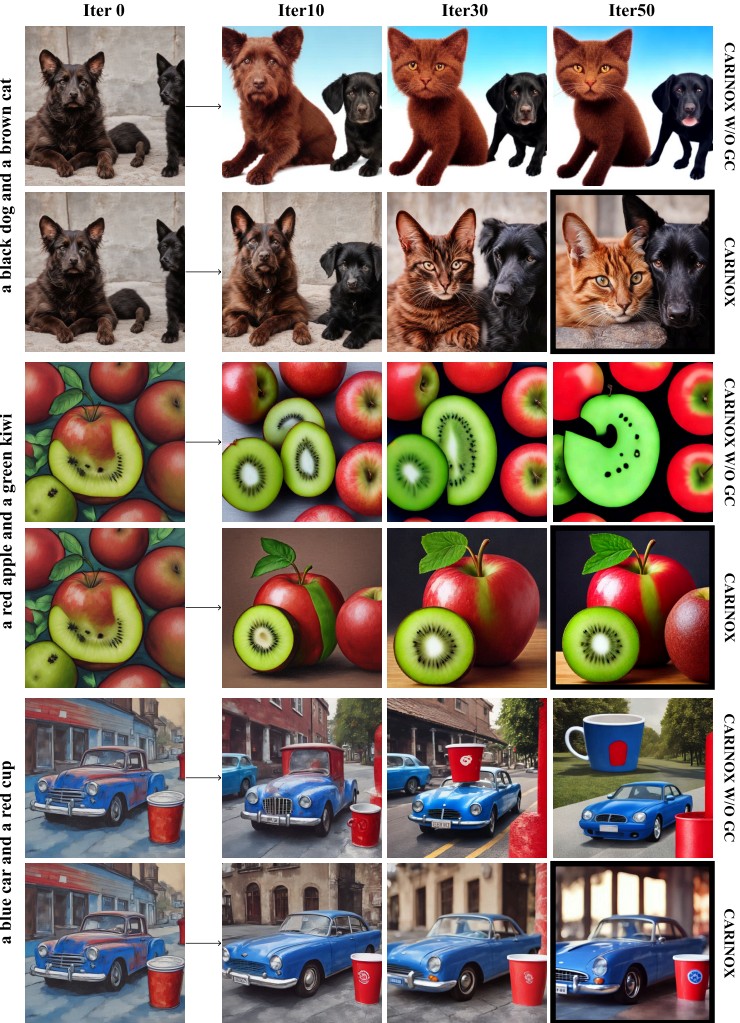

Figure 7: **Effect of Multi-Clip on Multi-Backward Optimization.** Without gradient clipping (top), dominant rewards distort updates: in "black dog and brown cat" the animals appear waxy and anatomically implausible, and in "red apple and green kiwi" the fruit exhibits unnatural texture, shading, and saturation. With Multi-Clip (bottom), each reward is balanced, preventing distributional drift and producing outputs that are both compositionally faithful and photo-realistic.

Figure 7 highlights the consequences of omitting this step. In the first example, the prompt "a black dog and a brown cat" leads to strong alignment of color and entities, but without clipping, the outputs drift toward *unrealistic, waxy, and anatomically implausible animals*. In the second case, the prompt "a red apple and a green kiwi" suffers a similar failure: although the objects and colors are correct, the fruit becomes highly *unnatural in texture, shading, and saturation*, far from realistic depictions. The third example with "a blue car and a red cup" shows the same pattern—objects are recognizable but appear distorted or cartoonish.

With **Multi-Clip** enabled, these issues are resolved. Each reward's gradient is scaled to contribute comparably, which stabilizes optimization, prevents distributional drift, and yields outputs that are *both compositionally correct and visually realistic*. In practice, this mechanism is essential for maintaining a balance between alignment fidelity and photo-realism, ensuring robust improvements across diverse prompts.

To provide a quantitative view of why Multi-Clip is needed, Table 8 reports both the mean and standard deviation of the per-reward gradient norms across iterations. Two patterns are consistent: (i) the scales remain mismatched throughout the trajectory (e.g., VQA stays the largest even at the end: Iter 50 mean 4.03 vs. HPS 0.22 and DA 0.16), so aggregation can remain dominated by a subset of rewards beyond the first few steps; and (ii) the standard deviations are large—especially early on (e.g., Iter 1: VQA std 112.12, ImageReward std 72.67)—indicating high variability and occasional extreme gradients that can abruptly steer the update direction. Multi-Clip addresses both by clipping each reward gradient independently before aggregation, preventing persistent or sporadic domination by any single reward.

| Iter | VQA Score | | ImageReward | | HPS | | DA Score | |
|---|---|---|---|---|---|---|---|---|
| | Mean | Std | Mean | Std | Mean | Std | Mean | Std |
| 1 | 39.09 | 112.12 | 11.52 | 72.67 | 2.37 | 3.36 | 6.27 | 15.19 |
| 10 | 15.25 | 71.11 | 2.79 | 11.84 | 0.77 | 1.08 | 1.37 | 3.28 |
| 20 | 8.70 | 31.43 | 1.36 | 4.71 | 0.45 | 0.57 | 0.64 | 1.70 |
| 30 | 6.36 | 30.54 | 0.82 | 5.85 | 0.32 | 0.39 | 0.33 | 0.92 |
| 40 | 4.85 | 14.26 | 0.51 | 1.77 | 0.26 | 0.29 | 0.21 | 0.54 |
| 50 | 4.03 | 10.64 | 0.39 | 1.89 | 0.22 | 0.29 | 0.16 | 0.53 |

Table 8: Mean and standard deviation of per-reward gradient norms $\|\nabla_\epsilon \mathcal{R}\|_2$ at selected optimization iterations (averaged over prompts/seeds used in the Multi-Clip analysis).

# F  Adaptive Reward Weighting (Ablation)

We include the following experiment as an ablation to assess whether replacing uniform reward weights ($\lambda_i=1$) with category-aware correlation-based weights improves optimization.

While a fixed combination of reward models provides a stable optimization framework, we evaluate a variant that assigns weights to each reward function based on its measured correlation with human preferences within each T2I-CompBench++ compositional category. The weight for each reward $\mathcal{R}_i$ is computed as:

$$w_i = \frac{4\rho_i}{\sum_j \rho_j},$$ (15)

where $\rho_i$ denotes the correlation coefficient of $\mathcal{R}_i$ with human evaluations for the corresponding category (Table 5). The weights are normalized to sum to 4 to keep the overall reward scale comparable to the uniform-weight setting. In this ablation, we use the ground-truth category labels provided by T2I-CompBench++ to select the category-specific weights.

Table 9 shows that adaptive weighting yields only marginal changes relative to uniform weighting. On SD-Turbo, AW slightly improves the mean score ($0.5542 \to 0.5595$), while on SDXL-Turbo it is essentially unchanged ($0.5554 \to 0.5539$), and on PixArt-$\alpha$ DMD it is again marginal ($0.5560 \to 0.5582$). Overall, these results suggest that the uniform weighting used in the main method is already a strong and robust default, and that correlation-based category-aware reweighting provides limited additional benefit under our current setup.

| Model | Color ↑ | Shape ↑ | Texture ↑ | 2D Spatial ↑ | 3D Spatial ↑ | Numeracy ↑ | Non-Spatial ↑ | Complex ↑ | Mean ↑ |
|---|---|---|---|---|---|---|---|---|---|
| (1) SD-Turbo | 0.5252 | 0.4434 | 0.4888 | 0.1881 | 0.3112 | 0.4914 | 0.3095 | 0.3349 | 0.3866 |
| (1) + CARINO | 0.8519 | 0.7336 | 0.8043 | 0.2437 | 0.3920 | 0.5903 | 0.3269 | 0.4906 | 0.5542 |
| (1) + CARINO (AW) | 0.8608 | 0.7430 | 0.8060 | 0.2411 | 0.4083 | 0.6036 | 0.3239 | 0.4896 | 0.5595 |
| (2) SDXL-Turbo | 0.5959 | 0.4038 | 0.5472 | 0.2303 | 0.3612 | 0.4863 | 0.3114 | 0.3430 | 0.4099 |
| (2) + CARINO | 0.8492 | 0.7203 | 0.7977 | 0.2858 | 0.4069 | 0.5835 | 0.3141 | 0.4859 | 0.5554 |
| (2) + CARINO (AW) | 0.8437 | 0.7238 | 0.8042 | 0.2767 | 0.3969 | 0.5848 | 0.3138 | 0.4877 | 0.5539 |
| (3) PixArt-$\alpha$ DMD | 0.4145 | 0.3487 | 0.3667 | 0.2213 | 0.3441 | 0.4896 | 0.3061 | 0.3466 | 0.3547 |
| (3) + CARINO | 0.8260 | 0.7528 | 0.7967 | 0.2620 | 0.4031 | 0.6144 | 0.3146 | 0.4782 | 0.5560 |
| (3) + CARINO (AW) | 0.8215 | 0.7562 | 0.8015 | 0.2770 | 0.4030 | 0.6098 | 0.3150 | 0.4813 | 0.5582 |

Table 9: Adaptive reward weighting (AW) ablation on T2I-CompBench++. AW replaces uniform weights with category-specific correlation-based weights, using the benchmark's ground-truth category labels to select weights for each prompt.

## G Human Evaluation Protocol

To assess alignment quality from a human perspective, we designed a four-level scoring scheme ranging from 0 to 3:

- **Score 0:** None of the objects described in the prompt are generated.

- **Score 1:** At least one object is present, but others are missing, severely deformed, or incorrectly generated.

- **Score 2:** All objects described in the prompt are present and recognizable, but attributes or relations (e.g., color, size, spatial layout, numeracy) may be incorrect or incomplete.

- **Score 3:** The image is fully consistent with the prompt: all objects are present, correctly rendered, and the specified attributes and relations are faithfully captured.

Seven annotators participated in the study, including four undergraduate and three master's students. Each annotator was provided with written instructions and example images corresponding to each score level to establish a consistent evaluation standard. The prompts were sampled from all eight compositional categories of *T2I-CompBench++*, and for each prompt, images from different methods were collected.

To avoid bias, images were presented in randomized order, with no information about which method or backbone produced them. Each annotator independently rated every image, ensuring multiple judgments per sample. The raw scores were then averaged across raters and normalized to the range $[0, 1]$ for reporting in the main paper. This protocol ensures both fairness and robustness of the human evaluation results.

## H Quality and Diversity Evaluation

Image quality and diversity remain central aspects of text–to–image generation, alongside compositional alignment. We therefore report Fréchet Inception Distance (FID)(Heusel et al., 2017), Density, and Coverage(Naeem et al., 2020) on the MS-COCO dataset (Lin et al., 2014). FID captures distributional distance from real images (lower is better), while Density and Coverage measure fidelity and diversity relative to the real distribution (higher is better).

Table 10 shows that CARINOX achieves competitive results on all three measures while providing substantial compositional improvements. In particular, Density and Coverage remain strong, confirming that the optimization framework preserves both realism and diversity of outputs. These results demonstrate that CARINOX delivers enhanced alignment without compromising overall generation quality.

| Model | FID ($\downarrow$) | Density ($\uparrow$) | Coverage ($\uparrow$) |
|---|---|---|---|
| SD v2.1 | 10.34 | 0.92 | 0.88 |
| + Attn-Exct | 10.35 | 0.91 | 0.88 |
| + InitNO | 7.38 | 0.93 | 0.91 |
| SD-Turbo | 8.09 | 0.70 | 0.99 |
| + ReNO | 10.12 | 0.97 | 0.99 |
| **+ CARINOX** | 12.93 | 0.91 | 0.97 |

Table 10: Quantitative comparison of quality and diversity between our proposed approach, CARINOX, and ReNO over the MS-COCO dataset. Lower FID values indicate better realism, while higher Density and Coverage values suggest better fidelity and diversity, respectively. While CARINOX provides a significant improvement in compositional generation, the degradation in quality and diversity is minimal.

# I Time and Memory Usage Analysis

## I.1 Measured Runtime and VRAM

We evaluate the computational efficiency of CARINOX by measuring runtime and VRAM usage on three backbones: PixArt-$\alpha$, SD-Turbo, and SDXL-Turbo. All measurements are obtained on an NVIDIA H100 GPU.

| Model | VRAM (GB) | Time (s) |
|---|---|---|
| (1) SD-Turbo | 10 | 0.15 |
| (1) + ReNO | 15 | 20 |
| (1) + CARINOX | 33 | 60 |
| (2) SDXL-Turbo | 16 | 0.25 |
| (2) + ReNO | 21 | 30 |
| (2) + CARINOX | 40 | 70 |
| (3) PixArt-$\alpha$ DMD | 21 | 0.12 |
| (3) + ReNO | 25 | 25 |
| (3) + CARINOX | 43 | 65 |

Table 11: Comparison of computation time and VRAM usage of CARINOX and ReNO over three different backbones.

As shown in Table 11, both CARINOX and ReNO introduce additional overhead compared to the raw backbones, but differ in their requirements due to the complexity of their optimization pipelines. CARINOX incurs higher runtime and VRAM usage than ReNO due to its multi-reward optimization and best-of-$N$ exploration. For example, on SD-Turbo, VRAM increases from $15\,\text{GB}$ (ReNO) to $33\,\text{GB}$ (CARINOX), and runtime increases from $20\,\text{s}$ to $60\,\text{s}$. Similar trends hold on PixArt-$\alpha$ and SDXL-Turbo.

The additional overhead in CARINOX stems from two factors: (i) *multi-reward optimization*, where each iteration evaluates multiple reward models and backpropagates their gradients, and (ii) *best-of-$N$ exploration*, where optimization is repeated independently for multiple noise seeds and the best result is selected. In particular, some of our rewards rely on heavier vision-language or VQA-style models, which are more demanding in both VRAM and runtime; using them jointly amplifies the per-iteration cost. Importantly, CARINOX is modular with respect to both components: if future work provides reward models with lower memory/latency or comparable quality at reduced cost, our framework can directly benefit by swapping in these scorers without changing the optimization procedure. Similarly, advances in seed exploration (e.g., more sample-efficient initialization search that achieves the same best-of-$N$ gains with fewer candidates) would reduce the number of optimized seeds required, lowering total runtime proportionally.

While CARINOX is more resource-intensive, the demands remain within the range of modern GPUs and are justified by the substantial performance gains achieved across benchmarks. This analysis illustrates the trade-off between computational cost and alignment quality, highlighting the importance of efficient reward integration in inference-time optimization.

### I.2 NFE-Matched Comparison

To address concerns about unequal inference-time compute, we additionally report NFE-matched comparisons, where NFE denotes the number of generator evaluations. For multi-step diffusion models, we set NFE $= K$ where $K$ is the number of denoising steps (we use $K = 50$ throughout). For single-step backbones (SD-Turbo, SDXL-Turbo, and PixArt-$\alpha$ DMD), a single forward generation corresponds to NFE $= 1$. Iterative noise-optimization methods (ReNO and CARINO) therefore use NFE $= T$, where $T$ is the number of optimization iterations. For CARINOX, the total compute scales as NFE $= N \times T$, where $N$ is the number of explored seeds and $T$ is the number of optimization iterations per seed. Under a fixed budget of NFE $= 50$, we report CARINO (50 iterations) and a compute-matched CARINOX configuration with $N = 2$ and $T = 25$ ($2 \times 25$). We also include our main CARINOX setting ($N = 5$, $T = 50$, NFE $= 250$) to illustrate the gains from additional inference-time scaling beyond the fixed-budget regime.

| Model | NFE | Color ↑ | Shape ↑ | Texture ↑ | 2D Spatial ↑ | 3D Spatial ↑ | Numeracy ↑ | Non-Spatial ↑ | Complex ↑ | Mean ↑ |
|---|---|---|---|---|---|---|---|---|---|---|
| SD v1.4 | 50 | 0.3765 | 0.3576 | 0.4156 | 0.1246 | 0.3030 | 0.4461 | 0.3079 | 0.3080 | 0.3299 |
| SD v2.1 | 50 | 0.5065 | 0.4221 | 0.4922 | 0.1342 | 0.3230 | 0.4579 | 0.3127 | 0.3386 | 0.3734 |
| SDXL | 50 | 0.5879 | 0.4687 | 0.5299 | 0.2133 | 0.3566 | 0.4988 | 0.3119 | 0.3237 | 0.4114 |
| PixArt-$\alpha$-ft | 50 | 0.6690 | 0.4927 | 0.6477 | 0.2064 | 0.3901 | 0.5058 | 0.3197 | 0.3433 | 0.4468 |
| Structured + SD v2.1 | 50 | 0.4990 | 0.4218 | 0.4900 | 0.1386 | 0.3224 | 0.4550 | 0.3111 | 0.3355 | 0.3717 |
| Attn-Exct + SD v2.1 | 50 | 0.6400 | 0.4517 | 0.5963 | 0.1455 | 0.3222 | 0.4550 | 0.3111 | 0.3355 | 0.4072 |
| InitNO + SD v2.1 | 50 | 0.7038 | 0.4694 | 0.5212 | 0.2027 | 0.3524 | 0.4892 | 0.3105 | 0.3574 | 0.4258 |
| (1) SD-Turbo | 1 | 0.5252 | 0.4434 | 0.4888 | 0.1881 | 0.3112 | 0.4914 | 0.3095 | 0.3349 | 0.3866 |
| (1) + ReNO | 50 | 0.7800 | 0.6200 | 0.7500 | 0.2200 | 0.3800 | 0.5700 | 0.3200 | 0.4800 | 0.5150 |
| (1) + CARINO ($T$=50) | 50 | 0.8519 | 0.7336 | 0.8043 | 0.2437 | 0.3920 | 0.5903 | 0.3269 | 0.4906 | 0.5542 |
| (1) + CARINOX ($N$=2, $T$=25) | 50 | 0.8383 | 0.7216 | 0.7940 | 0.2413 | 0.4015 | 0.6011 | 0.3264 | 0.4867 | 0.5531 |
| (1) + CARINOX ($N$=5, $T$=50) | 250 | 0.8633 | 0.7609 | 0.8229 | 0.2588 | 0.4155 | 0.6248 | 0.3372 | 0.5041 | 0.5734 |
| (2) SDXL-Turbo | 1 | 0.5959 | 0.4038 | 0.5472 | 0.2303 | 0.3612 | 0.4863 | 0.3114 | 0.3430 | 0.4099 |
| (2) + ReNO | 50 | 0.7800 | 0.6000 | 0.7400 | 0.2600 | 0.3900 | 0.5600 | 0.3100 | 0.4700 | 0.5137 |
| (2) + CARINO ($T$=50) | 50 | 0.8492 | 0.7203 | 0.7977 | 0.2858 | 0.4069 | 0.5835 | 0.3141 | 0.4859 | 0.5554 |
| (2) + CARINOX ($N$=2, $T$=25) | 50 | 0.8430 | 0.7141 | 0.7838 | 0.2951 | 0.4076 | 0.5803 | 0.3138 | 0.4880 | 0.5531 |
| (2) + CARINOX ($N$=5, $T$=50) | 250 | 0.8697 | 0.7482 | 0.8270 | 0.3010 | 0.4117 | 0.5992 | 0.3232 | 0.4922 | 0.5715 |
| (3) PixArt-$\alpha$ DMD | 1 | 0.4145 | 0.3487 | 0.3667 | 0.2213 | 0.3441 | 0.4896 | 0.3061 | 0.3466 | 0.3547 |
| (3) + ReNO | 50 | 0.6400 | 0.5700 | 0.7200 | 0.2500 | 0.3900 | 0.5600 | 0.3100 | 0.4600 | 0.4875 |
| (3) + CARINO ($T$=50) | 50 | 0.8260 | 0.7528 | 0.7967 | 0.2620 | 0.4031 | 0.6144 | 0.3146 | 0.4782 | 0.5560 |
| (3) + CARINOX ($N$=2, $T$=25) | 50 | 0.8175 | 0.7463 | 0.7803 | 0.2989 | 0.4019 | 0.5960 | 0.3139 | 0.4697 | 0.5522 |
| (3) + CARINOX ($N$=5, $T$=50) | 250 | 0.8545 | 0.7721 | 0.8076 | 0.3272 | 0.4164 | 0.6295 | 0.3256 | 0.4878 | 0.5776 |

Table 12: NFE-matched evaluation on T2I-CompBench++. For multi-step diffusion models and their inference-time variants, we report results at $K = 50$ denoising steps (NFE $= 50$). For single-step backbones, we report the raw generation reference (NFE $= 1$), a fixed compute budget setting (NFE $= 50$), and our main inference-time scaling setting (NFE $= 250$).

Table 12 indicates that the benefits of our approach are not tied to using a larger compute budget. In the NFE $= 50$ regime, **CARINO** and **CARINOX ($N$=2, $T$=25)** are the top-performing entries among the methods reported: they outperform the multi-step diffusion backbones (SD v1.4/v2.1/SDXL/PixArt-$\alpha$-ft), the inference-time attention methods on SD v2.1 (Structured Diffusion and Attend-and-Excite), the multi-step noise optimizer InitNO, and the single-step baseline ReNO. This shows that even when we match the evaluation budget to the standard 50-step setting, optimizing the single-step generator with our reward combination yields stronger compositional alignment than both multi-step generation and prior inference-time optimization strategies.

## J  Effect of Norm-Guided Regularization (Ablation)

We include this experiment as an ablation to isolate the impact of the norm-guided regularization term used in CARINOX. In this setting, we remove the regularization term from the optimization objective and keep all other components unchanged (same rewards, Multi-Clip, learning rate, number of iterations, and seed selection protocol).

Figure 8 visualizes the optimization trajectory at several iterations. Without regularization (top rows), the optimized samples progressively exhibit high-frequency noise, color saturation, and noticeable artifacts, indicating a distributional shift away from the generator's latent prior. This degradation becomes more pronounced at later iterations, even when the prompt-level attributes or entities appear partially satisfied.

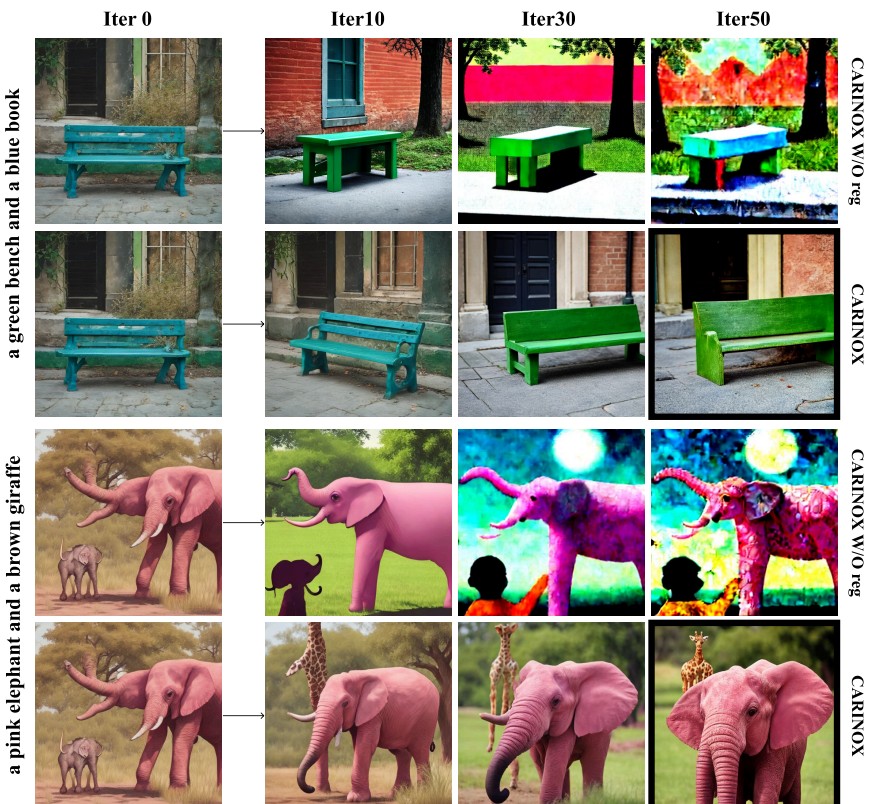

Figure 8: Ablation of the norm-guided regularization term. We show intermediate results across optimization iterations for two prompts. Without regularization (top rows), optimization increasingly drifts toward noisy and artifact-prone outputs (distributional shift). With regularization (bottom rows), optimization remains visually stable while improving alignment.

In contrast, the full CARINOX setting (bottom rows) preserves visual realism throughout optimization while still improving compositional alignment, suggesting that the norm constraint plays a practical role in stabilizing optimization and preventing drift into unrealistic regions of the image manifold.

## K   Pseudo code for Noise Optimization and Exploration

To provide a clearer understanding of our method, we present the pseudocode outlining the key steps of our initial noise optimization and seed exploration pipeline. This includes the gradient-based refinement of the initial noise using adaptive reward weighting, as well as the best-of-N seed selection strategy.

Algorithm 1 details the noise optimization process, where the initial noise is iteratively refined based on reward gradients while ensuring stability through multi-backward computation, gradient clipping, and latent space regularization. Furthermore, it describes the seed exploration approach, where multiple noise initializations are optimized in parallel, and the final selection is determined based on the highest reward score.

## L   Alternative Benchmark: GenEval

To further validate our method, we evaluate on the *GenEval* benchmark, which measures compositional alignment across categories such as single/multi-object generation, counting, color attribution, and spatial positioning. Results in Table 13 show that **CARINOX** achieves competitive or superior performance across both SD-Turbo and SDXL-Turbo backbones. In particular, it delivers strong improvements in color

---

**Algorithm 1** CARINOX: Reward-Guided Noise Optimization and Exploration

---

**Require:** $p$ (prompt), $G_\theta$ (One-Step T2I Model), $S_{1\dots N}$ (random seeds), $\mathcal{R}_{1\dots M}$ (reward functions), $K(\cdot)$ (noise regularizer), $T$ (iterations), $\eta$ (learning rate), $\tau$ (grad clip), $\gamma$ (regularization strength)

1: Sample $N$ initial noise vectors $\{\epsilon_1^0, \dots, \epsilon_N^0\} \sim \mathcal{N}(0, I)$          ▷ initialize Gaussian seeds
2: **for** $i = 1$ to $N$ **do**          ▷ exploration across multiple seeds
3:      Initialize best score $R_i^* \leftarrow -\infty$
4:      **for** $t = 0$ to $T - 1$ **do**          ▷ gradient ascent for each seed
5:          Generate image $I_i^t = G_\theta(\epsilon_i^t, p)$
6:          Initialize $\nabla_\epsilon \leftarrow 0$
7:          **for** $j = 1$ to $M$ **do**          ▷ loop over reward models
8:             Evaluate reward $r_j^t = \mathcal{R}_j(I_i^t, p)$
9:             Compute gradient $\nabla_\epsilon^j = \nabla_{\epsilon_i^t} [\mathcal{R}_j(I_i^t, p) + \gamma K(\epsilon_i^t)]$
10:            Clip gradient $\nabla_\epsilon^j \leftarrow \text{GradClip}(\nabla_\epsilon^j, \tau)$
11:            Accumulate $\nabla_\epsilon \leftarrow \nabla_\epsilon + \nabla_\epsilon^j$
12:          **end for**
13:          Compute total reward $R_i^t = \sum_{j=1}^M r_j^t$          ▷ composite reward for current step
14:          **if** $R_i^t > R_i^*$ **then**
15:            $R_i^* \leftarrow R_i^t, \quad I_i^* \leftarrow I_i^t$          ▷ update best image for this seed
16:          **end if**
17:          Update noise $\epsilon_i^{t+1} = \epsilon_i^t + \eta \cdot \nabla_\epsilon$          ▷ gradient ascent step
18:      **end for**
19:      Store final image $I_i = I_i^*$
20: **end for**
21: **return** $I^* = \arg\max_{I_i} \mathcal{R}(I_i, p)$          ▷ return best image across seeds

---

attribution and overall mean scores, matching or surpassing SOTA baselines including ReNO and large-scale commercial systems. These findings confirm that CARINOX generalizes well beyond the primary benchmarks used in the main paper.

| Model | Single ↑ | Two ↑ | Counting ↑ | Colors ↑ | Position ↑ | Color Attribution ↑ | Mean ↑ |
|---|---|---|---|---|---|---|---|
| SD v2.1 | 0.98 | 0.51 | 0.44 | 0.85 | 0.07 | 0.17 | 0.50 |
| SDXL | 0.98 | 0.74 | 0.39 | 0.85 | 0.15 | 0.23 | 0.56 |
| DALL-E 2 | 0.94 | 0.66 | 0.49 | 0.77 | 0.10 | 0.19 | 0.53 |
| DALL-E 3 | 0.96 | **0.87** | 0.47 | 0.83 | **0.43** | 0.45 | 0.67 |
| SD3 (8B) | 0.98 | 0.84 | 0.66 | 0.74 | 0.40 | 0.43 | **0.68** |
| (1) SD-Turbo | 0.99 | 0.51 | 0.38 | 0.85 | 0.07 | 0.14 | 0.49 |
| (1) + ReNO | **1.00** | 0.82 | 0.60 | 0.88 | 0.12 | 0.33 | 0.62 |
| (1) + CARINO | **1.00** | 0.84 | 0.53 | 0.85 | 0.12 | 0.40 | 0.62 |
| **(1) + CARINOX** | **1.00** | 0.86 | 0.54 | **0.90** | 0.13 | **0.48** | 0.65 |
| (2) SDXL-Turbo | **1.00** | 0.66 | 0.45 | 0.84 | 0.09 | 0.20 | 0.54 |
| (2) + ReNO | **1.00** | 0.84 | **0.68** | 0.90 | 0.13 | 0.35 | 0.65 |
| (2) + CARINO | **1.00** | 0.86 | 0.65 | 0.88 | 0.10 | 0.43 | 0.65 |
| **(2) + CARINOX** | **1.00** | 0.86 | 0.66 | **0.90** | 0.16 | **0.48** | **0.68** |

Table 13: Quantitative results on GenEval benchmark for different categories using two different backbones. For each category, the best value is bold, and the second-best value is underlined.

## M   Additional Qualitative Examples

To complement the main results, we provide additional qualitative comparisons between CARINOX and baseline methods. These examples further demonstrate the robustness of our approach across diverse compositional prompts, highlighting its ability to preserve both alignment fidelity and visual quality.

## Color Samples

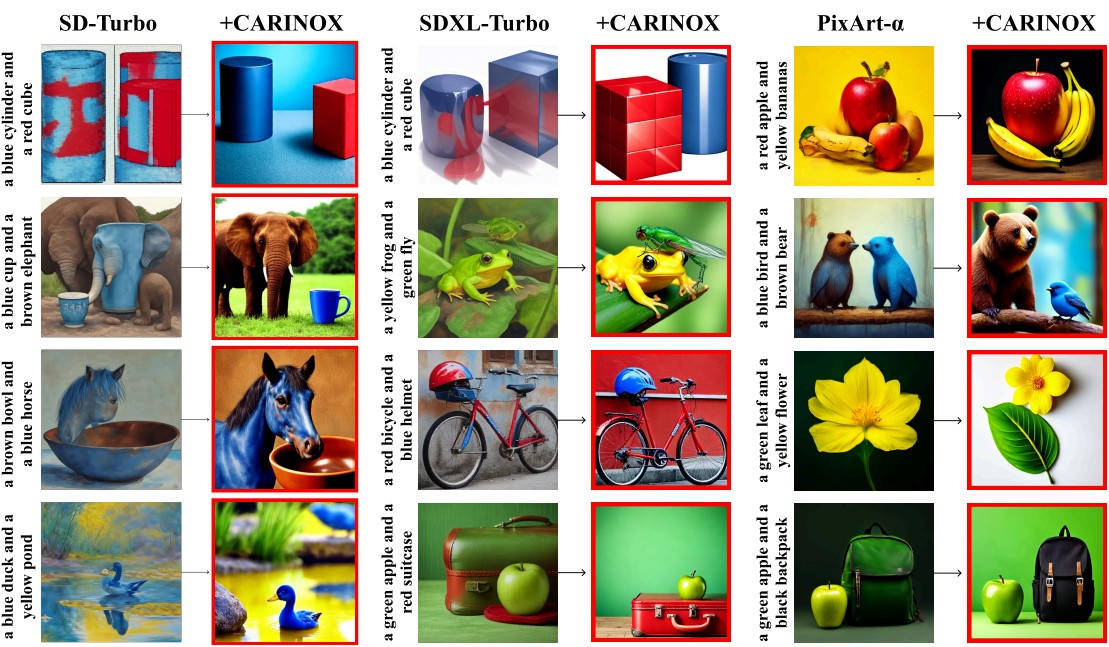

Figure 9: Qualitative examples for **color**. CARINOX adheres closely to specified colors and object–color bindings.

## Shape Samples

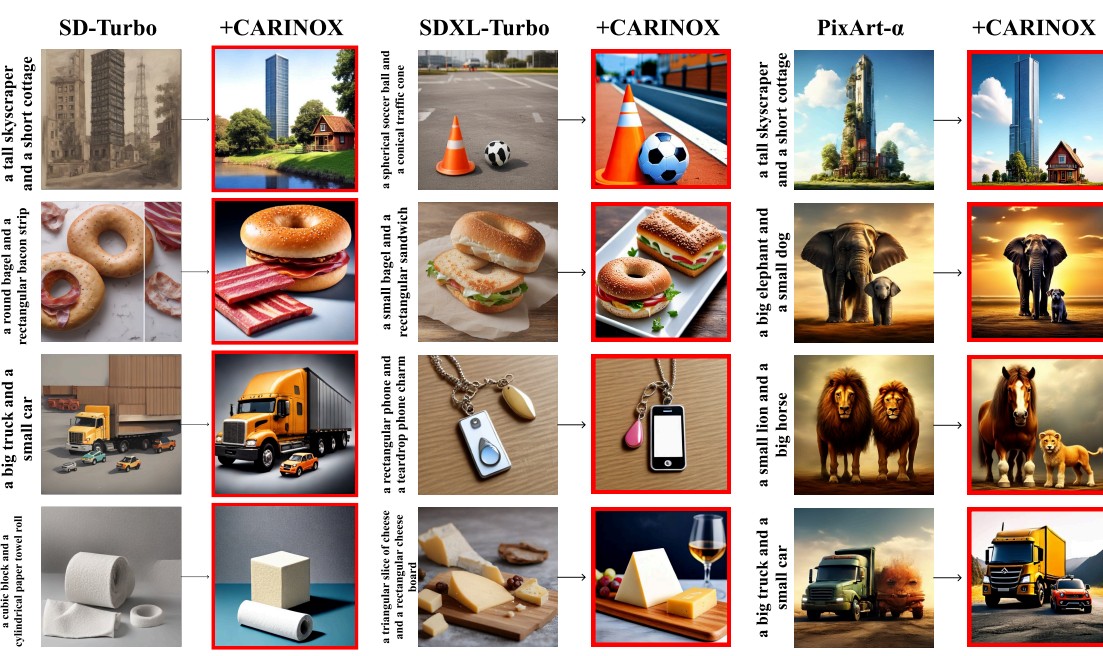

Figure 10: Qualitative examples for **shape**. CARINOX better preserves geometric structure and shape-specific attributes under compositional prompts.

## Texture Samples

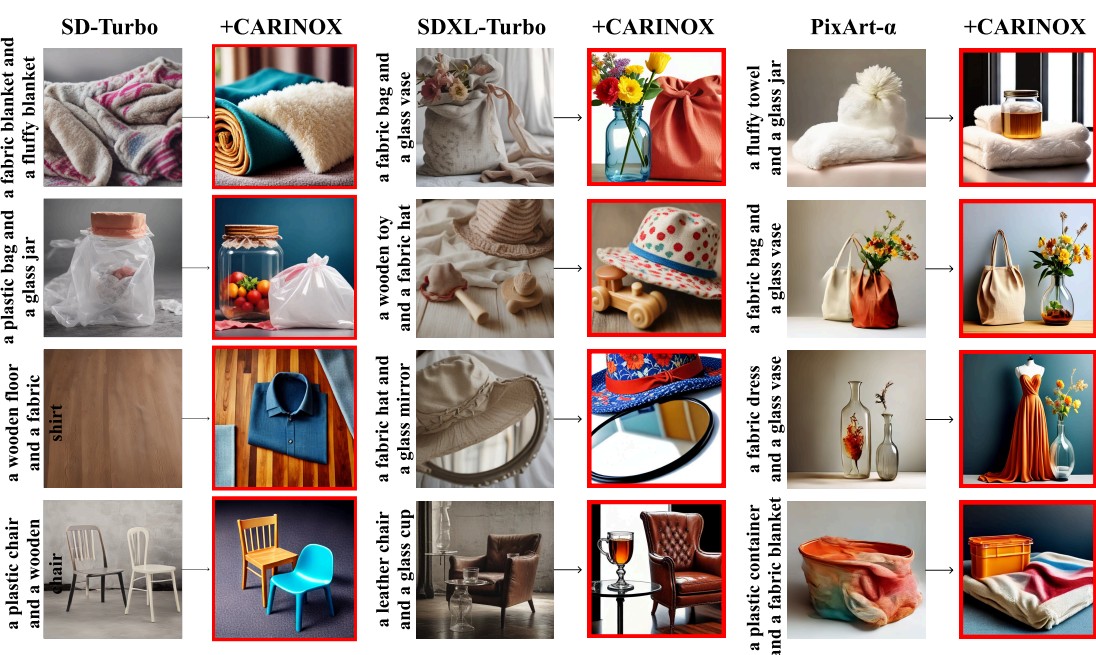

Figure 11: Qualitative examples for **texture**. CARINOX captures fine-grained surface patterns and material attributes more reliably.

## 2D Spatial Samples

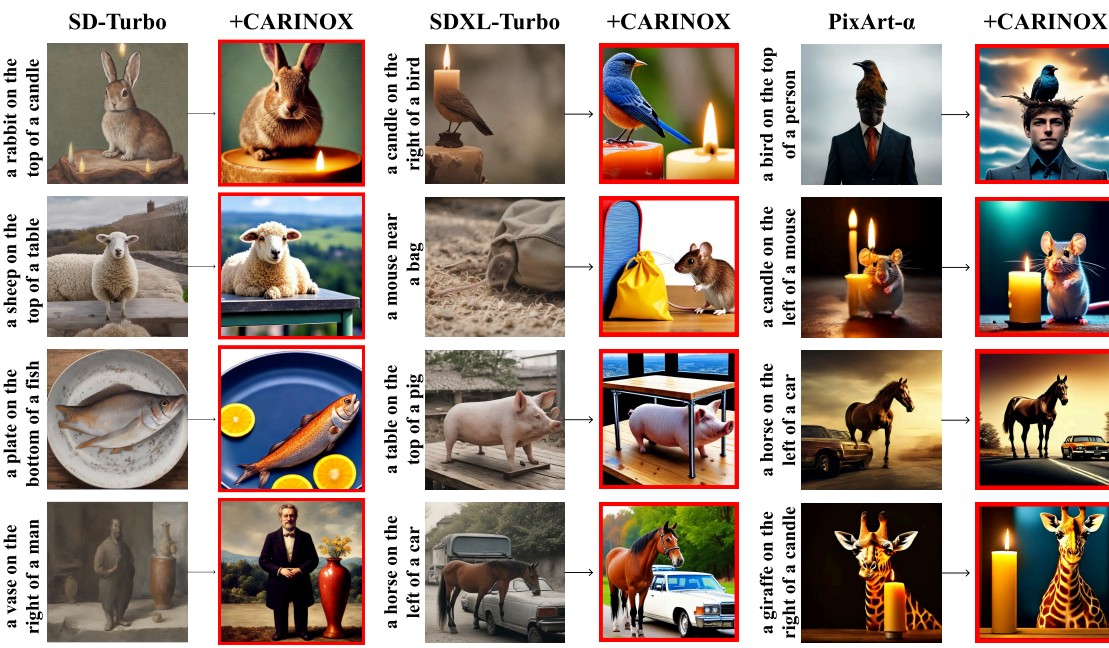

Figure 12: Qualitative examples for **2D spatial relations**. CARINOX produces layouts that more faithfully respect relative in-plane positions compared to baselines.

# 3D Spatial Samples

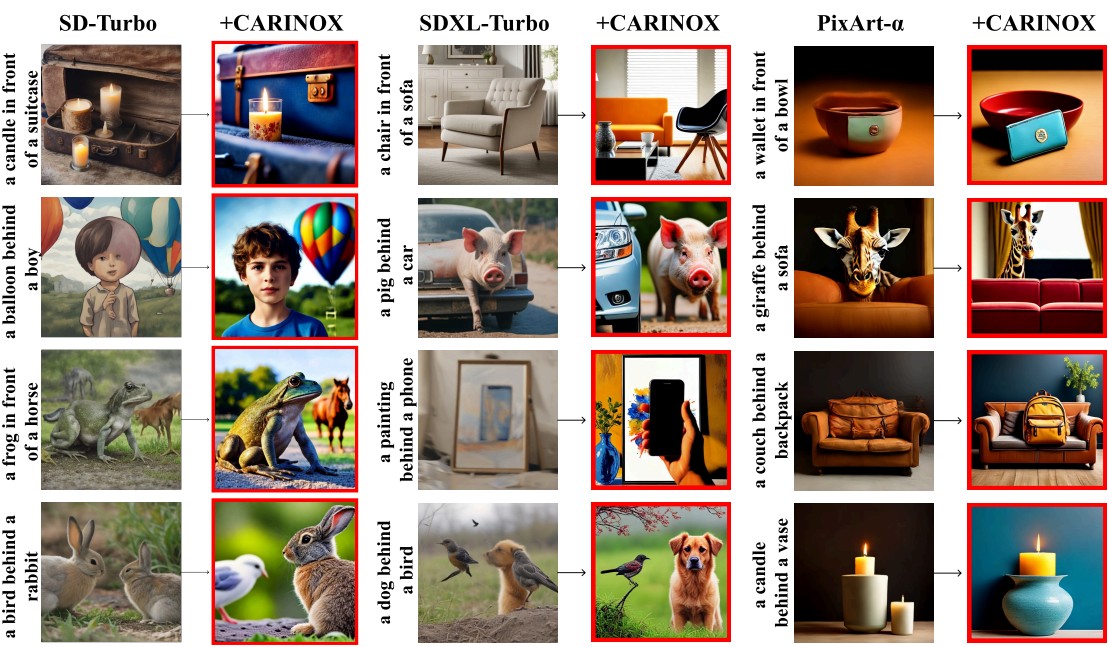

Figure 13: Qualitative examples for **3D spatial relations**. CARINOX better preserves depth and front–back/top–bottom relationships.

# Numeracy Samples

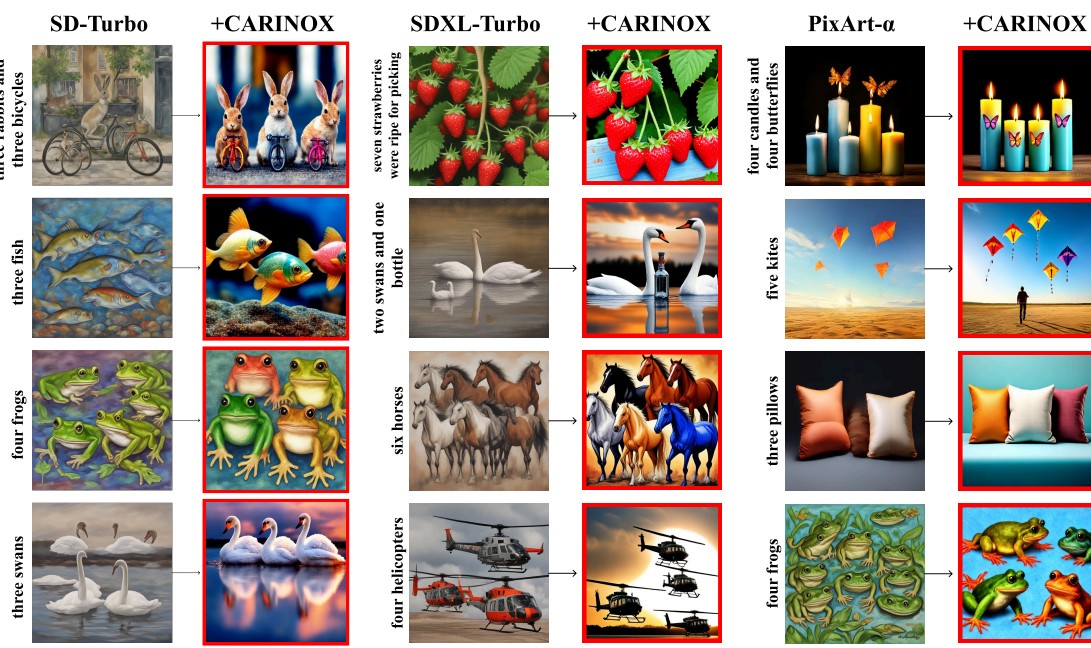

Figure 14: Qualitative examples for **numeracy**. CARINOX matches object counts and distributions more accurately than baselines.

## HRS Samples

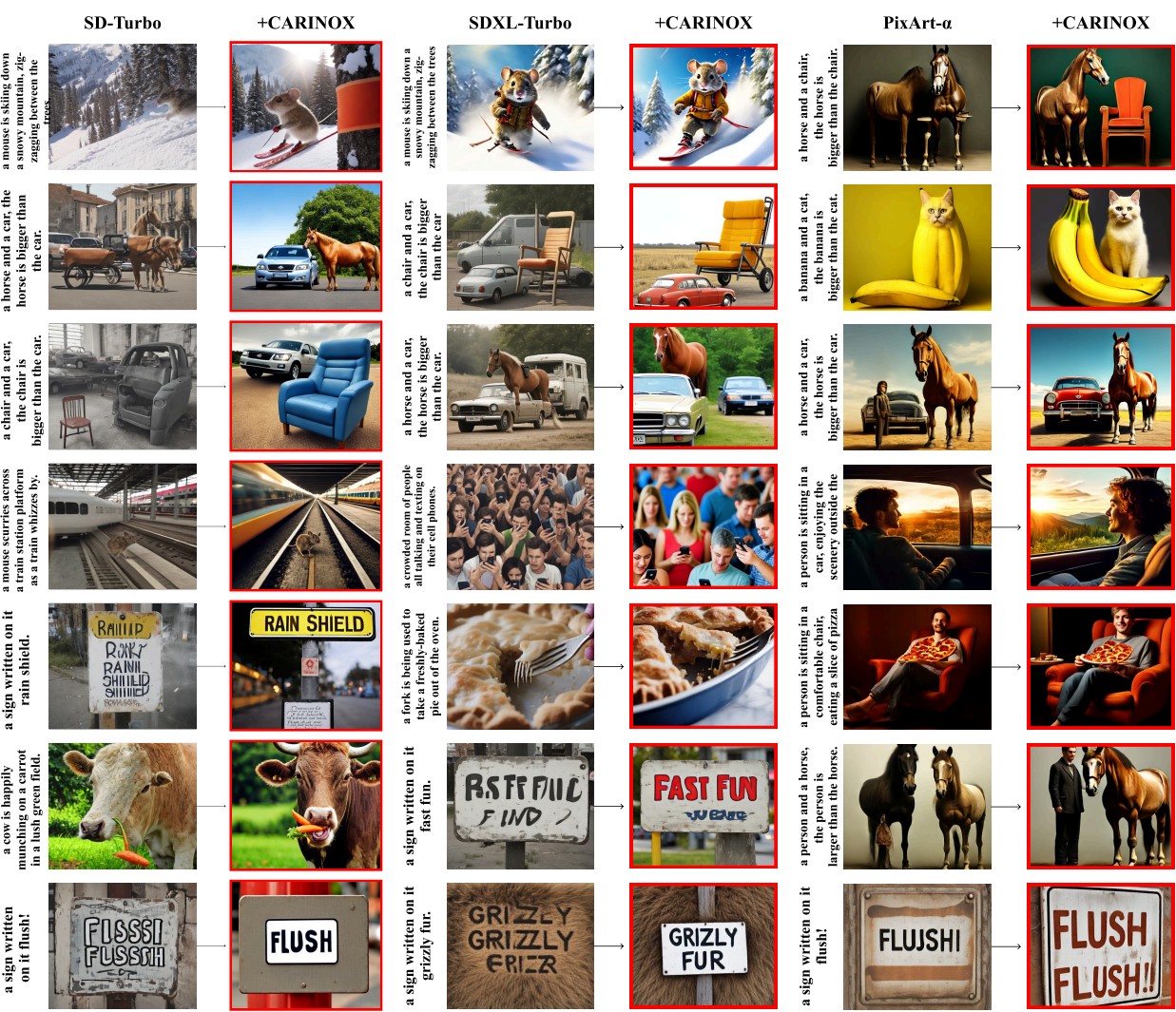

Figure 15: Additional qualitative results on the HRS benchmark. Examples show that CARINOX consistently improves compositional faithfulness over baseline models by correcting object relations, attributes, and text rendering.

