# OpenReview forum: "CARINOX: Inference-time Scaling with Category-Aware Reward-based Initial Noise Optimization and Exploration"
_TMLR — Accepted by TMLR_

### Review · Reviewer_r4G5 · 2025-10-18

**Summary Of Contributions:**

This paper presents **CARINOX**, an inference-time framework that improves compositional text-to-image generation by combining gradient-based initial noise optimization with discrete exploration across multiple candidate seeds. The method employs multiple reward functions to optimize for different aspects of composition. Comprehensive evaluations are presented on T2I-CompBench++ and HRS benchmarks and a subsidiary evaluation on GenEval, across multiple single-step diffusion backbones, which show substantial increase in performance over previous methods. The approach combines exploration diversity with optimization precision through best-of-N selection, iterative refinement, and multi-backward gradient computation.


**Key Strengths:**

- Clear and smooth flow paper writing, well written paper
- Unified framework addressing limitations of optimization-only and exploration-only methods
- Reward selection based on correlation study with human preferences following logical research protocol
- Per-reward gradient clipping mechanism is ingenious, preventing a single reward's dominance. Discussions in the paper providing reasoning for the mechanism is appropriate and welcomed.
- Regularization formulation is well grounded in theory and empirical evaluations
- Extensive qualitative and quantitative evaluation including standard benchmarks comparing against appropriate baselines, including appropriate derived conclusions and discussions on the findings
- Transparent reporting and empirical evaluation of limitations

**Key Weaknesses:**

- Substantial computational overhead (33GB VRAM, 60s generation time) undermines single-step model advantages

**Additional Comments:**

N/A

**Audience:**

Yes

**Audience Explanation:**

This paper would be of interest for researchers studying generative models, which would benefit from the ideas of improving inference time performance without fine-tuning.

**Claims And Evidence:**

Yes

**Claims Explanation:**

The claims made in the paper are backed by appropriate evidence. Comprehensive qualitative and quantitative evaluations have been presented across 3 standard benchmarks, as well as several competing baselines. The selection of the reward functions and their use is done using a well-thought research approach involving a human study. Subsidiary evaluations as well as detailed experimental setups and analyses have been presented in the paper. Barring few parameter selection criteria, the choices taken in terms of framework architecture, supporting parameters, choice of gradient clipping and regularization have been clearly justified and reasoned in the paper. Ablation studies on independent parts of the method's modules provide additional insights into the effectiveness of a combined method over standalone approaches. The limitations of the method are transparently presented with empirical evaluations, offering directions for future work.

**Requested Changes:**

### **Weaknesses :**

***Major*** -
- The computational cost of the method and time taken for generation is quite significant, especially when compared to the base single-step generation mode. This undermines the essence of single step T2I image generators which were developed to reduce both time and computation over their multi-step counterparts, while delivering non-compromising quality. But since the method utilizes ~33GB of VRAM and upto ~60s of generation time, this defeats the purpose of single-step generation models with this inference time methodology, and their multi-step counterparts which are generally better in performance vs time may be used
- While the limitation discussion and transparency is appreciated, an evaluation on the benchmarks comparing the [method + single step t2i models] vs multi-step models (without CARINOX) may be fruitful to see in terms of performance vs time, if by taking similar computational requirements the multi-step models perform in terms of generation time (I acknowledge that this was was partially represented on GenEval & T2ICompBench++ comparison). As I understand standard multi-step diffusion models have a typical generation time of 5-10s on an L40/A100 GPU. Please provide an appropriate comparison/discussion here

***Minor*** :
- An analysis of how many gradients are actually clipped to show quantitative effectiveness of the gradient clipping may be fruitful (it will also help identify which reward function is susceptible to gradient vanishing or explosion). A statistical evaluation could be added to offer more insight
- In Equation 9, which depicts the final constraint optimization equation, the paper does not specify how is  **γ** chosen, is it to be done through training or is it a hyperparameter to be tuned?
-An adaptive method could also explore different weight assignments to the rewards based on suitability to the prompt, wherein reward functions are weighted on prompt semantics. An ablation study on reward coefficients (**λ**) could be helpful here. The authors could offer an explanation on why λ values were all set to 1 and why is it an optimal solution
- I would advise some fruitful results from the appendix such as the GenEval analysis to be moved to the main content along with other additional suggestions from above
- It would be helpful to include exact GPU model and its TFLOP performance in the 'Time and Memory Usage Analysis' section. This would give a better idea of the actual computational performance

---

> ### Author Response · Authors · 2025-12-22
> **Response to Reviewer r4G5**
>
> Thank you for the thoughtful review and for highlighting both strengths and the main limitation around computational overhead. We respond point-by-point below.
>
> ---
>
> **Major concern 1: Runtime/VRAM overhead vs single-step advantage**
> We agree that CARINOX increases runtime and VRAM compared to the raw single-step backbone. As clarified in Appendix I.1, this overhead comes primarily from two factors: (i) multi-reward optimization, where each iteration evaluates multiple reward models and backpropagates their gradients, and (ii) best-of-N exploration, where the optimization is repeated for multiple noise initializations and the best result is selected. In our current setup, some reward models are relatively heavy (e.g., VLM/VQA-style scorers), and evaluating them jointly increases both VRAM and runtime. Importantly, CARINOX is modular with respect to both components: if future reward models provide comparable alignment signals at lower cost, or if exploration becomes more sample-efficient (fewer optimized seeds for the same final performance), our framework would directly benefit without changing the optimization procedure. We also include an equal-NFE comparison in Appendix I.2 to contextualize performance under a constrained compute budget.
>
> **Major concern 2: Comparison to multi-step models (performance vs time)**
> We appreciate this suggestion. Our main benchmark tables already include strong multi-step diffusion models as reference baselines (SD v1.4, SD v2.1, SDXL, PixArt-α-ft), and also include inference-time methods applied on SD v2.1 (Structured Diffusion, Attend-and-Excite, InitNO). This enables a direct comparison between (a) CARINO/CARINOX on single-step backbones and (b) multi-step diffusion models without CARINOX. In the revision, we make this comparison more explicit in the discussion, highlighting that CARINO/CARINOX not only improves over single-step baselines, but also surpasses these multi-step baselines and their inference-time variants in compositional alignment.
>
> **Minor concern 1: Quantitative analysis of gradient clipping**
> We add a quantitative per-reward gradient-norm analysis across optimization iterations in Appendix E (Table 8). This shows which reward gradients dominate at different stages and supports why per-reward clipping is necessary for stable multi-reward optimization.
>
> **Minor concern 2: Choice of λ in Eq. 9 and adaptive weighting**
> We clarify that in the main method all reward coefficients are fixed and set to λ=1 (no tuning or training). This avoids introducing extra hyperparameters and reflects our intent to let all selected rewards contribute equally, especially since different rewards can have different impacts during optimization. In response to your suggestion, we also include an ablation in Appendix F that tests correlation-based category-aware adaptive weighting as an alternative λ assignment; the impact is marginal relative to uniform weighting, supporting the default design.
>
> **Minor concern 3: Bringing appendix results into the main paper**
> We appreciate the suggestion. In the revision, we ensure the most essential evidence supporting the core claims is clearly presented in the main paper, while keeping supplementary analyses in the appendix for completeness.
>
> **Minor concern 4: GPU details**
> We specify that all runtime/VRAM measurements are obtained on an NVIDIA H100 GPU (Appendix I.1), and include this detail alongside the cost analysis for clearer interpretation and reproducibility.

---

> ### Comment · Action_Editor_xzJK · 2026-01-13
> **Final recommendation**
>
> Hi Reviewer,
>
> we are 1 week behind the schedule. Could you please submit your official recommendation ASAP Thanks?
>
> AE

---

> > ### Comment · Reviewer_r4G5 · 2026-01-16
> > **Official Recommendation**
> >
> > The revisions made have satisfactorily addressed my points and concerns. I appreciate the authors' effort to meet each of them appropriately.
> >
> > I recommend that the paper be accepted.

---

### Review · Reviewer_w6sc · 2025-10-23

**Summary Of Contributions:**

### **Summary**

This paper introduces **CARINOX**, an inference-time scaling framework that enhances text-to-image diffusion models by combining noise exploration and gradient-based noise optimization under multiple reward functions. It also proposes a **correlation-guided reward selection strategy** to identify complementary reward models that best align with human judgment. Experiments on T2I-CompBench++ and HRS benchmarks show that CARINOX significantly improves compositional alignment and semantic accuracy across backbones.

---
### **Strengths**:
Clear presentation: The paper is well written with clear structure, rich discussions, and an especially deep analysis of the Correlation-Guided Reward Combination Selection methodology.

Comprehensive evaluation: Systematic experiments across multiple diffusion backbones and diverse reward functions effectively demonstrate the robustness of the proposed approach.

----
### **Weaknesses**:
**Incomplete literature coverage**: The paper overlooks several related works, such as SMC-based methods (e.g., DAS [R1], RBF [R2], Psi-Sampler [R3]), initial noise optimization (e.g., ORIGEN [R4]), continuous latent optimization (e.g., DOODL [R5], D-Flow [R6]), and flow-based models like FLUX-schnell.

**Unfair computational comparison**: The evaluation uses different NFEs for baselines; fair comparisons should fix total NFE as done in prior literature.

**Limited ablation depth**: Additional studies could explore alternative exploration strategies (e.g., sampling N candidates and selecting top-K, or progressive pruning under equal NFE budgets) to better analyze efficiency–performance trade-offs.

---
### **References**
1. Kim et al., Test-time Alignment of Diffusion Models without Reward Over-optimization. ICLR 2025. (https://arxiv.org/abs/2501.05803)
2. Kim et al., Inference-Time Scaling for Flow Models via Stochastic Generation and Rollover Budget Forcing. NeurIPS 2025. (https://arxiv.org/abs/2503.19385)
3. Yoon et al., Psi-Sampler: Initial Particle Sampling for SMC-Based Inference-Time Reward Alignment in Score Models. NeurIPS 2025. (https://arxiv.org/abs/2506.01320)
4. Min et al., ORIGEN: Zero-shot 3D orientation grounding for text-to-image generation. NeurIPS 2025. (https://arxiv.org/abs/2503.22194)
5. Wallace et al., End-to-End Diffusion Latent Optimization Improves Classifier Guidance. CVPR 2023. (https://arxiv.org/abs/2303.13703)
6. Ben-Hamu et al., D-Flow: Differentiating through Flows for Controlled Generation. ICML 2024.(https://arxiv.org/abs/2402.14017)

**Audience:**

Yes

**Audience Explanation:**

Yes; the findings would interest TMLR readers working on diffusion models, inference-time optimization, and reward-guided generation.

**Claims And Evidence:**

Yes

**Claims Explanation:**

Mostly yes; the experimental evidence is convincing and clearly supports the main claims, though fairness regarding NFE differences should be better justified.

**Requested Changes:**

### **Critical (required for acceptance)**
**1. Ensure fair comparison by equalizing NFE across baselines.** The paper should either (a) align total NFEs across all baselines or (b) explicitly justify the differences in computational budgets. Currently, CARINOX uses up to 250 NFEs (5 seeds × 50 iterations), whereas CARINO and ReNO use only 50. For a fair comparison, exploration-only baselines like CARINX should use a comparable total NFE budget (e.g., at least 50 or 250, matching the total computation of CARINOX), or the paper should clearly explain why unequal NFEs are acceptable. Providing this clarification would improve the credibility and fairness of the evaluation and align with practices in prior inference-time optimization literature.

**2. Expand the literature review to include missing related works.**
 The paper should acknowledge and discuss prior methods that address similar goals using alternative paradigms:

- SMC-based methods: DAS [R1], RBF [R2], Psi-Sampler [R3]

- Initial noise optimization: ORIGEN [R4]

- Continuous latent optimization: DOODL [R5], DFlow [R6]

-----
### **Recommended (would strengthen the work):**
 **3. Introduce a more elaborate ablation on exploration strategies.**
 For instance, test sampling N candidates and selecting top-K based on reward before optimization, or implement progressive pruning under equal NFE constraints. Such experiments would clarify how exploration efficiency and compositional accuracy trade off.

**4. Discuss computation–performance trade-offs more explicitly.**
 Quantify and visualize how alignment improvements scale with NFE or GPU time, providing clearer justification for the “5 seeds × 50 iterations” design choice.

**5. Clarify the role of norm-guided regularization.**
 Include an ablation removing this component (w/o regularization) to empirically verify its effect on stability and over-optimization prevention.

---

> ### Author Response · Authors · 2025-12-22
> **Response to Reviewer w6sc**
>
> Thank you for the detailed review and for identifying concrete ways to strengthen the paper. We address each point below.
>
> ---
>
> **1. Fair comparison under equal NFE (required)**
> We agree that compute-budget alignment is important for inference-time scaling methods. In the revision, we (i) explicitly define the number of function evaluations (NFE) used by each setting, and (ii) add an equal-budget comparison where methods are evaluated under the same total NFE. This is reported in Appendix I.2 (“Same NFE Comparison”), including a compute-matched CARINOX configuration (2 seeds × 25 iterations, NFE=50) alongside CARINO (50 iterations, NFE=50) and the other baselines, while the full CARINOX setting (5×50) is presented as a higher-budget inference-time scaling regime. We also link these findings to the measured runtime/VRAM numbers in Appendix I.1.
>
> **2. Expanded literature review (required)**
> We expanded the Related Works section to include the missing lines of work you listed, placed in the corresponding inference-time categories: SMC-based approaches (DAS, RBF, Psi-Sampler), initial-noise optimization (ORIGEN), and continuous latent/flow optimization (DOODL and D-Flow).
>
> **3. Exploration strategy ablations (recommended)**
> Thank you for the suggestion. CARINOX is modular with respect to exploration: if a more sample-efficient exploration strategy is developed, our framework can directly benefit by reducing the number of optimized seeds needed for the same final performance. For the specific suggestion of selecting top-K candidates before optimization (or progressive pruning), a key issue is that pre-optimization quality/reward is not always predictive of post-optimization success—some initializations can lead the optimizer to unfavorable trajectories despite looking good initially, as also discussed in Section 3 of the paper.
>
> **4. Computation–performance trade-offs (recommended)**
> We clarify this trade-off explicitly in the revision by (i) reporting measured runtime/VRAM (Appendix I.1) and (ii) adding the equal-NFE results (Appendix I.2) to show what is achievable under a constrained compute budget versus higher-budget scaling. We also frame CARINOX as a test-time scaling method where the compute budget is explicitly controlled via “seeds × iterations.”
>
> **5. Norm-guided regularization (recommended)**
> Thank you for pointing this out. We add a dedicated ablation that removes the norm-guided regularization term while keeping the rest of the optimization pipeline unchanged, to isolate its role in stability and distributional drift; the qualitative comparison is included in Appendix J (Figure 8). (Separately, to further clarify stability mechanisms, we also expand the Multi-Clip analysis with additional quantitative evidence in Appendix E.)
>
> We hope these changes address the required concerns (literature completeness and equal-budget comparison) and clarify the efficiency–performance trade-offs raised in the review.

---

### Review · Reviewer_Qjhm · 2025-12-08

**Summary Of Contributions:**

This paper addresses the persistent issue of compositional misalignment in text-to-image diffusion models, including failures in object relations, attributes, spatial layouts, and counts. Rather than modifying model weights, the authors propose CARINOX, an inference-time method tailored to single-step diffusion models. The method incorporates:

- Initial Noise Optimization: gradient ascent on the noise vector using a composite reward, which is feasible with the choice of single-step diffusion architectures that would avoid long backpropagation chains.

- Noise Exploration: sampling multiple noise seeds, optimizing each, and selecting the highest-reward image.

- Principled Reward Selection: a correlation analysis on T2I-CompBench++ that identifies four reward models (HPS, ImageReward, DA Score, VQA Score) that best align with human judgments.

The approach maintains optimization stability through per-reward gradient clipping and a χ-distribution–based norm regularizer that keeps noise vectors close to the latent prior.

**Strengths**

- The combination of exploration and optimization is well motivated. Exploration improves diversity, and optimization refines candidates to achieve stronger compositional alignment.

- The systematic reward selection is a key contribution. Instead of relying on ad-hoc choices, the authors evaluate rewards via Spearman correlation with human-annotated categories and select those that consistently correlate best.

- The method demonstrates strong empirical performance. It outperforms baselines such as ReNO, InitNO, and ImageSelect across multiple backbones and compositional categories on T2I-CompBench++ and HRS. The ablation studies are also informative and support the design choices.

**Potential Concerns**

Overall, the approach is well motivated and I appreciate the principled way of choosing the reward functions. A few minor limitations include:

- The caption of Table 4 seems incorrect: the table reports how various reward choices favor different aspects of compositionality, but the caption describes an ablation study for Figure 5.

- Optimization must be run for each noise seed, so inference time increases linearly with the number of seeds.

- The optimization component applies only to single-step diffusion models; using it with multi-step models would require backpropagating through many denoising steps, which leads to vanishing or exploding gradients.

**Audience:**

Yes

**Audience Explanation:**

Yes. The paper addresses a known limitation of text-to-image diffusion models and proposes an inference-time solution that avoids retraining. The approach is well motivated and demonstrates strong empirical performance.

Researchers working on diffusion models, controllability, and evaluation would likely find the work relevant. The systematic reward-selection procedure and the unified optimization–exploration framework could also be of interest to practitioners developing or assessing inference-time enhancement methods.

**Claims And Evidence:**

Yes

**Claims Explanation:**

Yes, the claims are generally well supported.

- The paper provides comparisons between exploration-only (CARINX), optimization-only (CARINO) variants, and the combined method (CARINOX), which demonstrates the benefit of integrating both components.

- Both qualitative examples (Figure 4) and quantitative evaluations (Table 2) on T2I-CompBench++ show that CARINOX outperforms baselines such as ReNO, InitNO, ImageSelect, etc. across multiple categories and backbones.

- The reward-selection procedure is justified through a Spearman correlation study (Table 5), where the four chosen rewards tend to show the strongest alignment with human-annotated scores.

- Ablation studies further validate the impact of iteration count, seed count, and individual reward components, supporting the methodological choices.

**Requested Changes:**

The caption in Table 4 should be corrected.

---

> ### Author Response · Authors · 2025-12-22
> **Response to Reviewer Qjhm**
>
> Thank you for the careful reading and for the positive assessment of our exploration+optimization framework and correlation-guided reward selection. We address your concerns point-by-point below.
>
> ---
>
> **1. Table 4 caption mismatch**
> You are right. We corrected the caption of Table 4 in the revised version.
>
> **2. Linear scaling with the number of explored seeds**
> We agree that exploration increases compute linearly with the number of seeds, since each seed is optimized independently. To make this trade-off explicit, we add (i) a brief breakdown of where the runtime/VRAM cost comes from (multi-reward optimization and best-of-N exploration) in Appendix I.1, and (ii) an equal-NFE comparison in Appendix I.2 that includes a constrained setting (e.g., fewer seeds / fewer total evaluations). This shows that CARINO and a compute-matched CARINOX configuration remain strong under a fixed budget, while the full setting provides additional gains.
>
> **3. Applicability to multi-step diffusion models**
> We agree that gradient-based optimization is currently most practical for single-step generators, because backpropagating through many denoising steps in standard multi-step diffusion is more costly and can suffer from vanishing/exploding gradients. This motivation is clarified in the revision, and we also note that multi-step diffusion models are included as reference baselines in our comparisons. Despite focusing the optimization component on single-step generators, CARINOX achieves strong empirical gains: it improves single-step backbones and surpasses a broad range of baselines in our evaluation, including multi-step diffusion models and inference-time methods applied to them, as well as single-step inference-time baselines such as ReNO and exploration-based methods.

---

> ### Comment · Action_Editor_xzJK · 2026-01-13
> **Final recommendation**
>
> Hi Reviewer,
>
> we are 1 week behind the schedule. Could you please submit your official recommendation ASAP Thanks?
>
> AE

---

### Author Response · Authors · 2025-12-22

Thank you to all reviewers for the time, careful consideration, and constructive feedback. We have uploaded a revised version of the paper that addresses the comments, and we have submitted point-by-point responses to each review. We appreciate the reviewers’ effort in helping us improve the work, and we look forward to the final decision.

---

### Decision · Action_Editor_xzJK · 2026-01-22

**Recommendation:** Accept as is

**Audience:**

Yes

**Audience Explanation:**

Optimizing text-to-image diffusion models remains a highly active research area, even when the focus is restricted to inference-time optimization.

**Claims And Evidence:**

Yes

**Claims Explanation:**

This paper claims to propose a novel framework for optimizing the diffusion initial noise at inference time, outperforming the state of the art. The method is evaluated on two benchmarks and demonstrates meaningful improvements. In addition, the paper empirically studies various reward functions within the proposed framework. All reviewers are satisfied with the empirical evaluation and the observed performance gains, which substantiate the submission’ claims.